# The holocephalan ratfish endoskeleton shares trabecular and areolar mineralization patterns, but not tesserae, with elasmobranchs little skate and catshark

Oghenevwogaga Joseph Atake[1], Fidji Berio[2], Melanie Debiais Thibaud[2], B Frank Eames[1]*

[1]Department of Anatomy, Physiology, and Pharmacology, University of Saskatchewan, Saskatoon, Canada; [2]Institut des Sciences de l'Evolution, University of Montpellier, CNRS, IRD, EPHE, Montpellier, France

## eLife Assessment

This **important** study significantly advances our understanding of the skeleton of cartilaginous fishes by using a range of state of the art and complementary approaches to compare the skeleton amongst three cartilagenous fishes (catshark, little skate and ratfish). The evidence presented is **compelling** and likely to impact several fields of study.

*For correspondence:
b.frank@usask.ca

Competing interest: The authors declare that no competing interests exist.

**Abstract** Specific character traits of mineralized endoskeletal tissues need to be clearly defined and comprehensively examined among extant chondrichthyans (elasmobranchs, such as sharks and skates, and holocephalans, such as chimaeras) to understand their evolution. For example, tiles of mineralized polygonal structures called tesserae occur at cartilage surfaces in chondrichthyans, but recent studies showing trabecular mineralization at elasmobranch cartilage surfaces suggest that tesserae are not as common as previously thought. Also, while areolar mineralized tissue in elasmobranchs is generally considered a unique, shared chondrichthyan feature, some chondrichthyan species demonstrate bone-like tissues in both a specific region of tesserae termed the cap zone and continuous (not tiled) mineralized neural arches. To clarify the distribution of specific endoskeletal features among extant chondrichthyans, adult skeletal tissues in a holocephalan chimaera (spotted ratfish) and two elasmobranchs (small-spotted catshark and little skate) were characterized using synchrotron radiation and desktop micro-CT imaging, and histological and immunofluorescent assays. Endoskeletal mineralization in the ratfish, catshark, and little skate varied both quantitively in tissue mineral density (TMD) and qualitatively in the morphology and localization of mineralized structures and tissues. For example, TMD of several skeletal elements was significantly lower in ratfish, compared to catshark and little skate. Trabecular and areolar mineralization were shared among these extant chondrichthyan species, but tesserae and bone-like tissues were not. Interestingly, three separate analyses argued that the adult chimaera endoskeleton has features of the embryonic little skate endoskeleton. Generally, this study proposes specific terminology for character states of the extant chondrichthyan endoskeleton and infers those states in ancestral chondrichthyans with reference to fossil data.

## Introduction

The skeleton of chondrichthyans, or cartilaginous fishes, has long fascinated evolutionary biologists (*Daniel, 1934*; *Janvier, 1996*; *Kölliker, 1860*; *Maisey, 2013*; *Ørvig, 1951*; *Smith and Hall, 1990*; *Wurmbach, 1932*). However, only in the past two decades has an active research community revealed the histological and molecular complexity of extant chondrichthyan skeletal tissues. Even so, these studies are limited in scope, focusing mostly on specific skeletal features of the Elasmobranchii subclass of extant Chondrichthyes, such as sharks, skates, and rays (*Atake et al., 2019*; *Dean et al., 2017*; *Eames et al., 2007*; *Johanson et al., 2013*; *Kemp and Westrin, 1979*; *Omelon et al., 2014*; *Porter et al., 2006*; *Seidel et al., 2016*), with only a few including the Holocephali subclass, such as chimaera (*Gillis et al., 2011*; *Herbert et al., 2022*; *Pears et al., 2020*; *Seidel et al., 2020*; *Smith et al., 2020*). Unfortunately, an in-depth description of the extant holocephalan endoskeleton is lacking, leading to a deficit in understanding skeletal tissue character states across extant chondrichthyans.

Like most vertebrates, skeletal mineralization in extant chondrichthyans occurs by incorporation of biominerals into extracellular compartments of skeletal tissues (*Golub, 2009*; *Lowenstam, 1981*; *Omelon et al., 2014*), giving rise to diverse mineralized tissue types in several anatomical sites of the chondrichthyan endoskeleton (*Table 1*). Traditionally recognized as a defining endoskeletal feature of Chondrichthyes, tesserae form a mineralization pattern consisting of an array of distinct, polygonal units at the surface of cartilages (*Kemp and Westrin, 1979*; *Maisey, 2013*; *Maisey et al., 2021*). In skates, however, a recently characterized trabecular mineralization pattern lies directly underneath the polygonal pattern of tesserae (*Atake et al., 2019*). In fact, the cartilage surface in several endoskeletal regions of skates and rays only contains the trabecular mineralization pattern (*Atake et al., 2019*; *Jayasankar et al., 2020*), suggesting that polygonal tesserae may not be as common as traditionally thought. Like polygonal tesserae, the little skate trabecular mineralization also forms an arrayed pattern, but the repeating units are in the form of trabecular struts interspersed by unmineralized regions, leading us to call them trabecular tesserae (*Atake et al., 2019*; *Atake and Eames, 2021*). Descriptions of a trabecular mineralization pattern in the endoskeleton of sharks or chimaeras are currently lacking, leaving open the possibility that this is a phenotype specific to the Batoidea superorder (including skates and rays) of the subclass Elasmobranchii. In some chimaeras, the surfaces of cartilage in the anterior fused vertebrae (synarcual) and chondrocranium have sheets of mineralization that abut one another (*Pears et al., 2020*; *Seidel et al., 2020*), but these do not have an arrayed pattern of repeating units, so perhaps they should not be called tesserae.

Recent evidence suggests that histological features of tesserae may vary according to the mineralization pattern. Traditional polygonal tesserae have distinct histological zones, with a cap zone superficial to a body zone near the surface of chondrichthyan cartilage (*Atake et al., 2019*; *Berio et al., 2021*; *Kemp and Westrin, 1979*; *Seidel et al., 2016*). While skates have polygonal tesserae, trabecular tesserae in little skate have very small or even absent cap zones overlying extensive body

**Table 1.** Character states of the mineralized chondrichthyan endoskeleton.

Traits do not apply to all regions of a given fish's skeleton.

| Character | State | Description |
|---|---|---|
| Mineralization patterns | Polygonal tesserae (PT) | Array of distinct polygonal units between the perichondrium and trabecular tesserae |
| | Trabecular tesserae (TT) | Array of distinct trabecular struts, often with hypermineralized spokes, below the perichondrium |
| | Non-tesseral trabeculae (NTT) | Non-arrayed, distinct trabecular struts, often with hypermineralized spokes, below the perichondrium |
| | Areolar (AR) | Continuous, compact sheets or rings in the notochordal sheath |
| | Bone-like tissue (BLT) | Continuous, compact regions between the perichondrium and a cartilaginous core, often in neural arches and cap zones |
| Histological features | Body zone (BZ) | Distinct region of mineralized tissue below the perichondrium, containing Col2, rounded cell lacunae, and low-cellularity spokes |
| | Cap zone (CZ) | Distinct region of mineralized tissue between the perichondrium and body zone, containing tightly wound Col1 fibers and elongated cell lacunae |
| | Bone-like tissue (BLT) | Mineralized tissue of tightly wound Col1 fibers and elongated cell lacunae connected by canaliculi-like channels |

zones (*Atake et al., 2019*). Additionally, specific regions of tesserae termed spokes are hypermineralized and often devoid of cells, and spokes appear to be a common feature of tesserae regardless of their mineralization pattern (*Atake et al., 2019*; *Pears et al., 2020*; *Seidel et al., 2020*; *Seidel et al., 2016*). Careful 3D spatial analyses are needed to correlate histological zones and mineralization patterns of tesserae.

Some chondrichthyans have mineralized endoskeletal tissues that have been characterized as bone-like (*Atake and Eames, 2021*). For example, the cap zone of tesserae displays bony features, like elongate cell lacunae in a mineralized ECM of densely packed type I collagen (Col1) fibers (*Atake et al., 2019*; *Kemp and Westrin, 1979*; *Seidel et al., 2017*; *Seidel et al., 2016*). Unlike the cap zone, the body zone displays typical features of hyaline cartilage, which is the main cartilage type in vertebrates (*Goldring et al., 2006*). The body zone consists of round chondrocyte lacunae embedded in a mineralized matrix abundant in loosely packed type II collagen (Col2) fibers and sulfated glycosaminoglycans (*Enault et al., 2015*; *Seidel et al., 2017*). As another example of bone-like tissue, several elasmobranch species have a non-tiled, mineralized perichondral tissue in their neural arches (*Atake et al., 2019*; *Berio et al., 2021*; *Bordat, 1987*; *Eames et al., 2007*; *Kemp and Westrin, 1979*; *Seidel et al., 2016*). The neural arch bone-like tissue demonstrates morphological and histological features of perichondral bone, including a compact mineralization pattern, elongate cell lacunae, canaliculi-like channels connecting adjacent lacunae, and tightly packed Col1 fibers (*Bordat, 1987*; *Eames et al., 2007*; *Kemp and Westrin, 1979*; *Peignoux-Deville et al., 1982*; *Rossert and Crombrugghe, 2002*; *Seidel et al., 2017*). Bone-like tissues might be a defining feature of extant chondrichthyans, but relevant data from chimaeras are needed.

Finally, areolar mineralized tissue is generally considered a unique feature of the extant chondrichthyan endoskeleton, characterized mostly in elasmobranchs as large sheets of mineralization in the notochordal sheath of the vertebral body (centrum; *Didier, 1995*; *Ridewood and MacBride, 1921*). Areolar mineralized tissue typically contains concentric lamellae of elongate lacunae in a fibrocartilage-like extracellular matrix and a compact mineralization pattern (*Atake et al., 2019*; *Criswell et al., 2017*; *Dean and Summers, 2006*; *Eames et al., 2007*). Mineralized centra are not thought to be common in chimaeras, but when present, they can appear as multiple mineralized rings in each vertebral segment, which contrasts with the grossly biconcave morphology of segmented centra in elasmobranchs (*Gadow and Abbott, 1895*).

With hopes of revealing shared and clade-specific features among chondrichthyans, here we clarify many of the unresolved issues highlighted above, proposing specific terminology for character states of the extant chondrichthyan endoskeleton (*Table 1*). Testing the hypothesis that extant chondrichthyans have homologous histological and morphological traits of endoskeletal mineralized tissues, adult tissues in several endoskeletal regions of the spotted ratfish *Hydrolagus colliei* (chimaera), small-spotted catshark *Scyliorhinus canicula* (shark), and little skate *Leucoraja erinacea* (skate) were characterized using synchrotron radiation (SR) and desktop micro-CT imaging, and histological and immunofluorescent assays. Tesseral mineralization, cap zones, and neural arch bone-like tissues were absent in the ratfish, but present in these elasmobranchs. Three separate analyses argued that the endoskeleton of this chimaera displays histological and morphological features of the embryonic little skate endoskeleton. While additional chondrichthyan species can bolster these findings, the data argue that trabecular and areolar mineralization patterns, but not tesserae, are shared among extant chondrichthyans. Considering fossil outgroups, we finish by inferring ancestral and derived character states of the chondrichthyan endoskeleton.

## Results

### Adult mineralized tissues in spotted ratfish had lower tissue mineral density (TMD) than small-spotted catshark or little skate

To assess whether chimaeras demonstrate skeletal mineralization in anatomical sites that are homologous to those of elasmobranchs, 3D rendering of micro-CT images of anterior regions of adult ratfish was qualitatively and quantitatively analyzed (*Figure 1A and B*). Micro-CT images revealed mineralized tissues in several endoskeletal regions, including neural arches and centra of segmented vertebrae (*Figure 1C and D*), skeletal elements in the pharyngeal skeleton (*Figure 1E*), and the fused vertebral synarcual (*Figure 1F and G*). Although elasmobranchs mineralize neural arches in their precaudal and

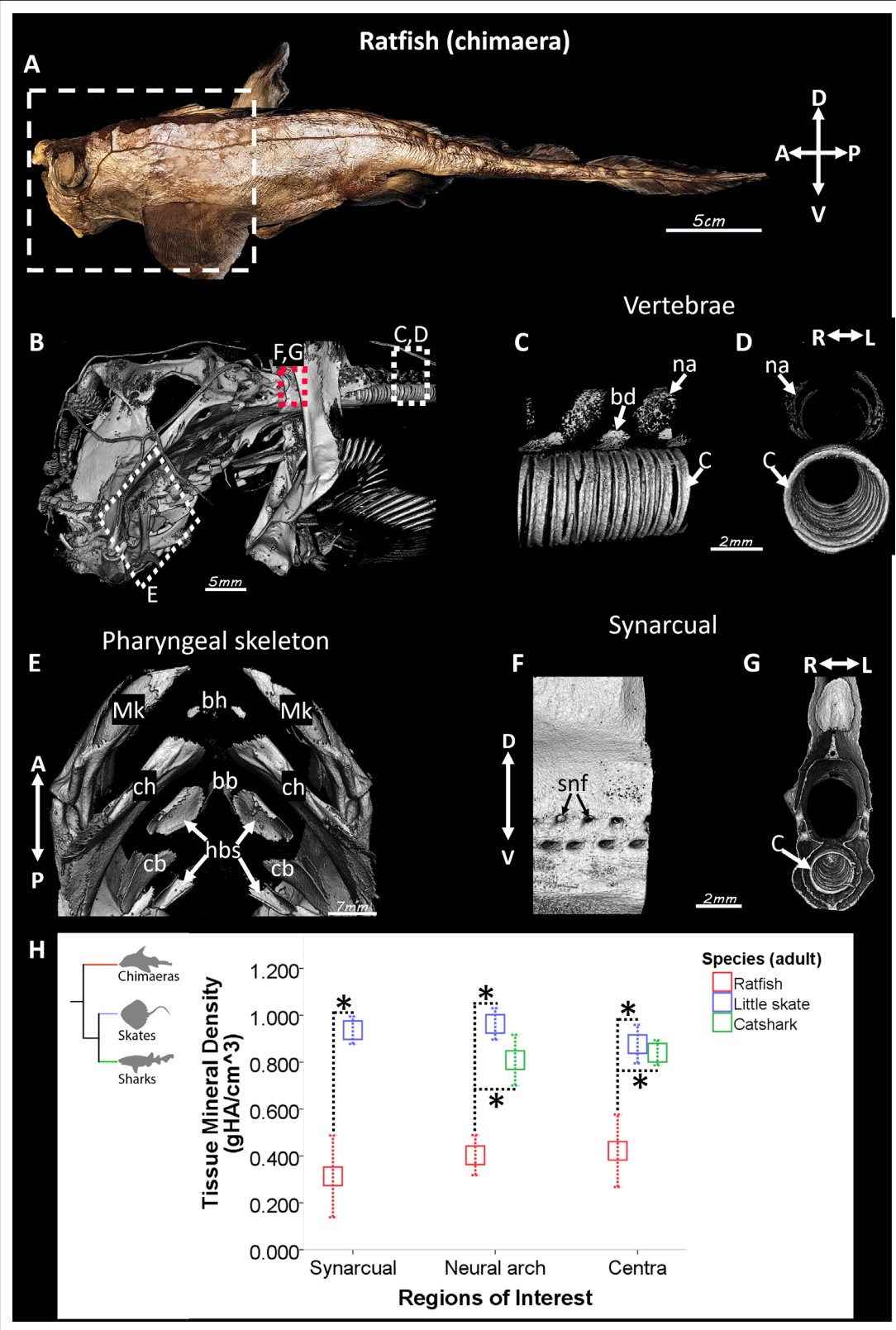

**Figure 1.** Mineralized tissues were abundant in the spotted ratfish, but they had lower tissue mineral density (TMD) compared to the little skate and the small-spotted catshark. (**A**) Photograph of the ratfish *Hydrolagus colliei* illustrating the anterior region that was micro-CT imaged. (**B–G**) Micro-CT 3D renderings of the anterior region of ratfish (**B**) showed mineralized tissues in neural arches, basidorsals, and centra in segmented vertebrae (**C, D**), in the ceratohyal and other skeletal elements in the pharyngeal skeleton (**E**), and in the fused vertebral synarcual (**F, G**). (**H**) Comparative quantitative analyses

*Figure 1 continued on next page*

*Figure 1 continued*

showed that TMD of the synarcual, neural arch, and centra in ratfish were significantly lower than TMDs of same regions in little skate and catshark. Abbreviations: A, anterior; bb, basibranchial; bd, basidorsal; bh, basihyal; c, centrum cb, ceratobranchial; ch, ceratohyal; D, dorsal; hbs, hypobranchials; L, left; Mk, Meckel's; na, neural arch; P, posterior; R, right; snf, spinal nerve foramina; V, ventral. * indicates statistically significant comparison (p<0.05). Scale bars: as indicated.

caudal vertebrae (*Atake et al., 2019*; *Berio et al., 2021*; *Eames et al., 2007*), mineralized neural arches were present mainly in precaudal vertebrae of ratfish (*Figure 1C and D*; data not shown). Mineralized skeletal elements in the pharyngeal skeleton included the basihyal, Meckel's, ceratohyal, basibranchial, hypobranchials, and ceratobranchial (*Figure 1E*). Spinal nerve foramina are common in the synarcual of cartilaginous fishes and were present in the ventral sides of the ratfish synarcual (*Figure 1F*; *Claeson, 2011*; *Johanson et al., 2015*; *Pears et al., 2020*). Unlike the synarcual in skates and other batoids, where mineralized centra are not discernible (*Atake et al., 2019*; *Claeson, 2011*), the synarcual in ratfish had distinct mineralized centra for its entire length (*Figure 1G*).

TMDs of the synarcual, neural arches, and centra in ratfish, catshark, and little skate were quantitated and compared, where possible, since sharks do not have a synarcual (*Figure 1H*). Because little skate does not have centra within the synarcual, centra in the ratfish synarcual were digitally segmented and excluded for synarcual TMD data. The average TMD of the ratfish synarcual was $0.31 \pm 0.11$ gHA/cm$^3$, significantly lower than the TMD of the little skate synarcual, which was $0.94 \pm 0.04$ gHA/cm$^3$ (*Figure 1H*; $p = 6.5 \times 10^{-4}$). Similarly, the TMD of ratfish neural arches ($0.40 \pm 0.05$ gHA/cm$^3$) was significantly lower than the TMD of neural arches of either catshark ($0.81 \pm 0.04$ gHA/cm$^3$; $p = 9 \times 10^{-6}$) or little skate ($0.96 \pm 0.04$ gHA/cm$^3$; $p = 4.5 \times 10^{-7}$; *Figure 1H*). The average TMD of ratfish centra ($0.42 \pm 0.10$ gHA/cm$^3$) was also significantly lower than the TMD of centra of either catshark ($0.84 \pm 0.02$ gHA/cm$^3$; $p = 1.1 \times 10^{-4}$) or little skate ($0.88 \pm 0.05$ gHA/cm$^3$; $p = 3.4 \times 10^{-5}$; *Figure 1H*). In ratfish, TMD of segmented centra within the synarcual was similar to that of centra in segmented vertebrae ($0.46 \pm 0.11$ gHA/cm$^3$). These analyses showed that the synarcual, neural arches, and centra in ratfish had significantly lower TMDs compared to homologous elements in catshark and little skate.

## Neural arches in ratfish lacked bone-like tissue

To clarify whether chimaeras have bone-like tissue like elasmobranchs, micro-CT and histological analyses of neural arches in segmented precaudal vertebrae of ratfish were compared to neural arches in segmented precaudal vertebrae of catshark and little skate. Mineralization of neural arches in ratfish was organized in a non-continuous pattern of small compact and trabecular regions (*Figure 2A*). This mineralization pattern in spotted ratfish contrasted with the continuous mineralization pattern displayed by the compact, bone-like tissue in catshark and little skate (*Figure 2B and C*; *Atake et al., 2019*; *Berio et al., 2021*). Furthermore, cross-section views demonstrated that mineralized tissues in ratfish were restricted to the medial and lateral periphery of the neural arches (*Figure 2D*). By contrast, mineralization of bone-like tissues in catshark and little skate extended much deeper in the neural arch (*Figure 2E and F*).

Histologically, Alizarin red staining of calcium showed that mineralized tissues in ratfish neural arches were much thinner and not continuous like bone-like tissues in catshark and little skate neural arches (*Figure 2G–I*). Alcian blue staining of sulfated proteoglycans marked cartilage in most neural arch regions in ratfish, while only the cartilaginous core regions of catshark and little skate neural arches were Alcian blue-positive (*Figure 2G–I*). The neural arch core in catshark was mostly unmineralized cartilage, while that of little skate was mineralized cartilage (*Figure 2H, I*). Birefringence of picrosirius-stained, tightly packed collagen fibers (likely reticular) was observed in muscles lateral to neural arches in all three species (*Figure 2J–L*). Birefringence was not observed in ratfish neural arches, whereas birefringence in neural arch bone-like tissues was observed in catshark and little skate (*Figure 2K and L*). Most regions (including unmineralized cartilage and mineralized tissue) in ratfish neural arches demonstrated Col2 immunofluorescence (*Figure 2J*). Only the cartilaginous core in neural arches of catshark and little skate demonstrated Col2 immunofluorescence (*Figure 2K and L*). These results showed that bone-like tissue was absent in neural arches of ratfish, and the non-continuous pattern of mineralized tissue in ratfish neural arches was mineralized cartilage.

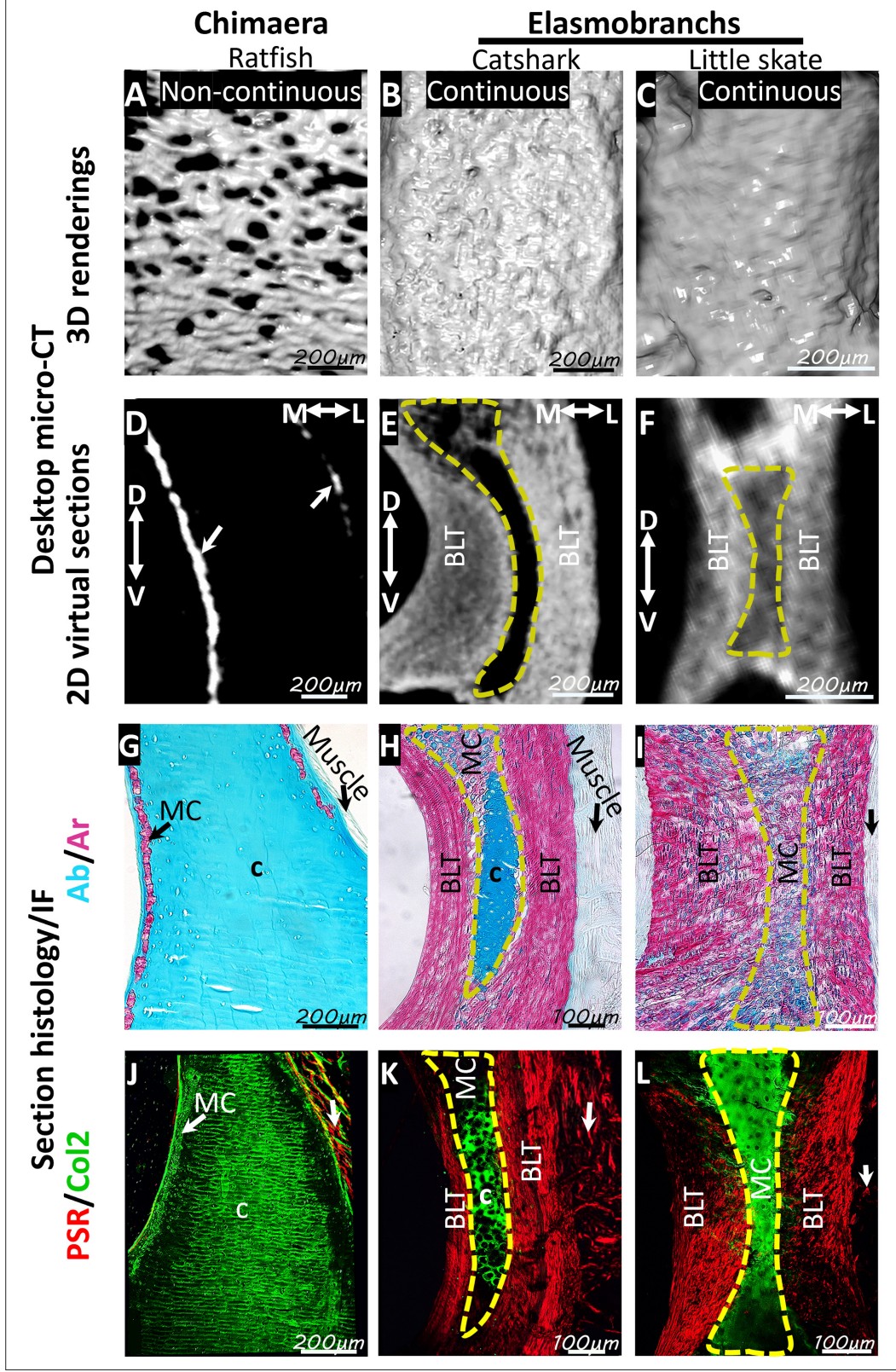

**Figure 2.** Ratfish lack neural arch bone-like tissue. (**A–C**) Micro-CT 3D renderings showed a non-continuous pattern of mineralized tissues in neural arches of segmented vertebrae in ratfish (**A**), compared to the continuous mineralization pattern of compact, bone-like tissue in catshark (**B**) and little skate (**C**). (**D–F**) Micro-CT 2D virtual slices showed that mineralized tissues (arrows) in ratfish neural arches were restricted to medial and lateral

*Figure 2 continued on next page*

*Figure 2 continued*

peripheries (**D**), whereas the bone-like tissue in catshark and little skate extended much deeper in the neural arch (**E, F**). (**G–I**) Alizarin red staining of histological sections confirmed micro-CT representation of mineralization patterns in ratfish (**G**), catshark (**H**), and little skate (**I**) neural arches, and Alcian blue revealed cartilaginous regions. (**J–L**) Picrosirius red staining did not reveal birefringent collagen fibers in ratfish neural arches (**J**), but such staining was abundant in bone-like tissues in catshark (**K**) and little skate (**L**). Birefringent fibers (likely reticular) were present in muscles (vertical arrows) on the lateral sides of neural arches in all species. Col2 immunostaining confirmed Alcian blue staining, identifying cartilage regions in all species. The neural arch core in catshark (**K**) and little skate (**L**) consisted of mostly unmineralized cartilage and mineralized cartilage, respectively. 2D, two-dimensional; 3D, three-dimensional; Ab, Alcian blue; Ar, Alizarin red; BLT, bone-like tissue; c, cartilage; Col2, collagen type 2; D, dorsal; IF, immunofluorescence; L, lateral; M, medial; MC, mineralized cartilage; PSR, picrosirius red; V, ventral. Scale bars: as indicated.

## Novel characterization of non-tesseral trabecular mineralization in ratfish and catshark showed the absence of a bone-like cap zone

To clarify whether tesserae are defining endoskeletal features of extant chondrichthyans, morphological patterns and histological features of mineralized tissues were characterized in ceratohyals of ratfish, caudal vertebrae of catshark, and precaudal vertebrae and hyomandibulae of little skate. Desktop micro-CT renderings of ratfish ceratohyals and catshark haemal arches revealed mineralized tissues with a trabecular pattern like that of trabecular tesserae in the vertebral neural spine of little skate (*Figure 3A–G*). However, the trabecular pattern in ratfish and catshark was not as clearly organized into arrays of repeating trabecular units like that of trabecular tesserae in little skate, and for this reason, trabecular patterns in ratfish and catshark were characterized as non-tesseral. In the hyomandibulae and other skeletal elements of little skate, including synarcual, propterygium, and vertebral transverse processes, a trabecular mineralization pattern occurred underneath the superficial polygonal tesseral pattern (*Figure 3D, H and H'*; *Atake et al., 2019*). Notably, many regions in ratfish (vertebrae, pharyngeal skeleton, and the synarcual) and catshark (vertebrae and ceratohyals) did not exhibit the polygonal mineralization pattern typical of tesserae (*Atake et al., 2019*; *Kemp and Westrin, 1979*; *Maisey, 2013*; *Marramà et al., 2019*; *Seidel et al., 2016*; *Wilga and Ferry, 2015*).

To assess whether trabecular mineralized tissues in ratfish, catshark, and little skate exhibit similar histological features, cross-sections of tissues with non-tesseral patterns were compared to those with tesseral patterns. Non-tesseral mineralized tissues in Alcian blue-positive perichondral regions in ratfish ceratohyals and catshark hemal arches stained with Alizarin red (*Figure 3I and J*; perichondrium marked by straight dashed line). The discrete organization of mineralized tissues and Alcian blue and Alizarin red staining patterns of non-tesseral mineralized tissues in ratfish and catshark were similar to the tesseral body zone in little skate (*Figure 3I–L*). However, Alcian-blue-positive cartilage separated Alizarin-red-positive non-tesseral mineralized tissues from overlying perichondrium in ratfish and catshark (*Figure 3I and J*). A bone-like cap zone just below the perichondrium was only present in tesseral mineralized tissues in little skate (*Figure 3K and L*; perichondrium marked by straight dashed line). The bone-like cap zone stained more intensely than the body zone with Alizarin red and was relatively smaller and larger in trabecular tesserae and polygonal tesserae, respectively (*Figure 3K and L*). Perichondrium of both non-tesseral and tesseral mineralized tissues stained with the general connective tissue dye fast green (*Figure 3M–P*). Matrix of the little skate bone-like cap zone was fast green-positive and had birefringent collagen fibers, but these staining patterns were not observed in non-tesseral mineralized tissues in ratfish and catshark (*Figure 3M–T*). Instead, regions just below the perichondrium of ratfish ceratohyals and catshark haemal arches stained for sulfated proteoglycans by Safranin O and were Col2-positive cartilage (*Figure 3M, N, Q and R*). Staining patterns of non-tesseral mineralized tissues in ratfish and catshark were generally consistent with those of mineralized cartilage in the tesseral body zone of little skate, although chondrocyte lacunae in these regions were not obvious in ratfish (*Figure 3I–T*). Therefore, histological features of trabecular mineralized tissues in ratfish ceratohyals and catshark haemal arches were similar to only the body zones of tesserae in little skate, which consisted of mineralized cartilage.

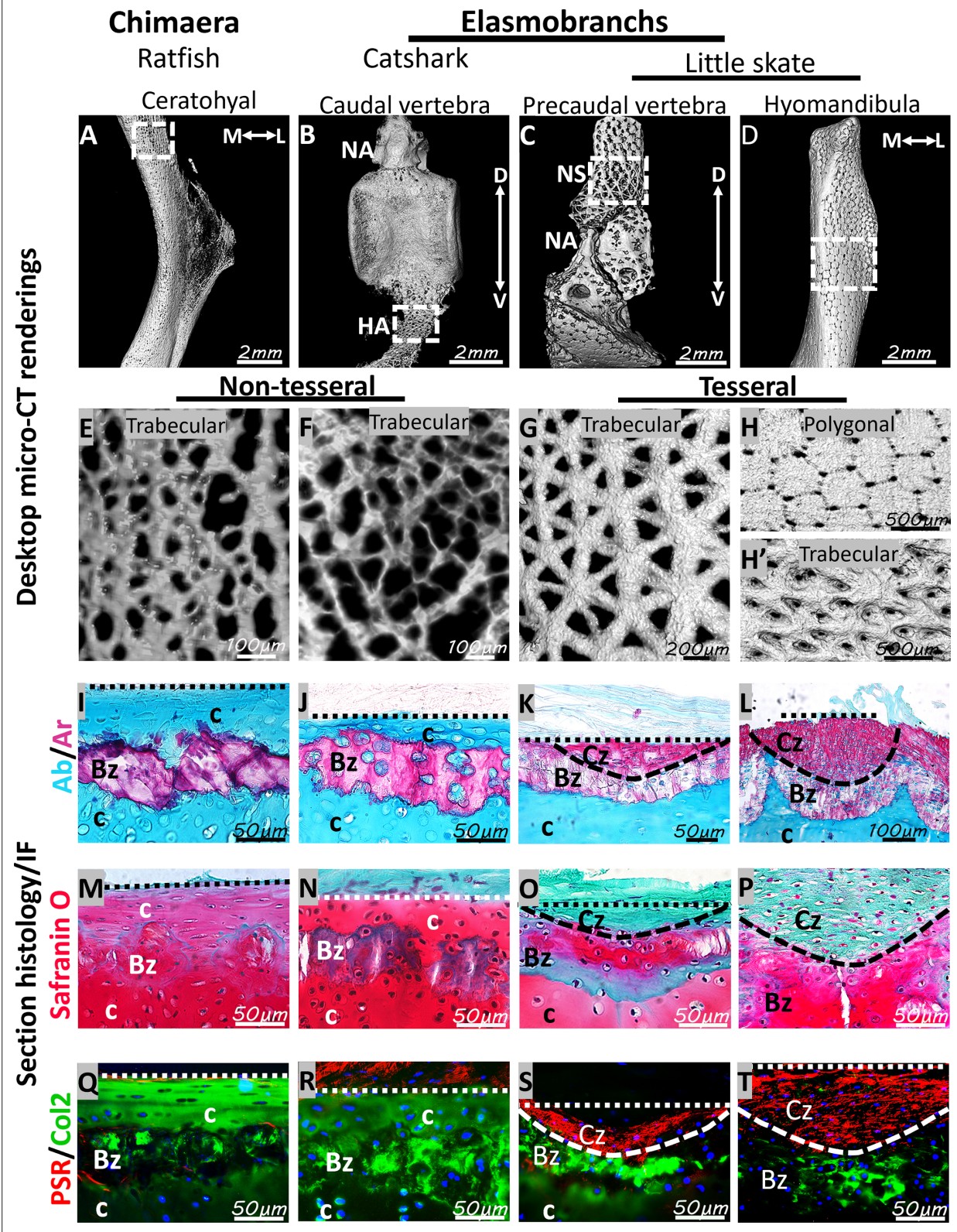

**Figure 3.** Ratfish, catshark, and little skate all showed trabecular mineralization, but ratfish and catshark did not have a bone-like cap zone. (**A–H**) Micro-CT renderings of the ratfish ceratohyal (**A, E**) and catshark caudal vertebra (**B, F**) demonstrated non-tesseral trabecular mineralization, while little skate hyomandibula had tesseral trabecular mineralization (**C, G**). Little skate precaudal vertebra had tesseral polygonal mineralization (**D, H**) overlying tesseral trabecular mineralization (**H'**). Regions for hi-mag views in panels E-H' are indicated in corresponding panels of A-D. (**I–P**) Alcian blue/Alizarin

*Figure 3 continued on next page*

*Figure 3 continued*

red and Safranin O/fast green staining of histological sections demonstrated that the body zone of non-tesseral or tesseral trabecular mineralized tissues in ratfish (**I, M**), catshark (**J, N**), and little skate (**K, L, O, P**) was mineralized cartilage, while the cap zone of tesseral polygonal mineralization was bone-like (**K, L, O, P**). (**Q–T**) Col2 immunostaining of adjacent sections to panels I-P confirmed cartilage regions in all species, while picrosirius red staining confirmed the bone-like tesseral cap zone in little skate. Ab, Alcian blue; Ar, Alizarin red; Bz, body zone; c, cartilage; Col2, collagen type 2; Cz, cap zone; D, dorsal; IF, immunofluorescence; L, lateral; M, medial; PSR, picrosirius red; NA, neural arch; NS, neural spine; V, ventral. Scale bars: as indicated.

## Spatial organization of histological features correlated with trabecular and polygonal mineralization patterns

Given that trabecular mineralized tissues typically have small or even absent cap zones, and polygonal tesserae typically have large cap zones, histology and micro-CT segmentation were used to see whether there was a correlation between histological features and mineralization patterns in ratfish and little skate. After demineralization, non-tesseral tissues in ratfish and tesseral tissues in little skate exhibited spokes, which are regions of very low cellularity in the body zone that often appeared as tears or sectioning artifacts and did not stain strongly with any Trichrome dye (*Figure 4A–C*). Non-spoke regions of the body zone stained either slightly blue for collagen fibers with Trichrome's aniline blue or magenta with Trichrome's acid fuchsin, and these staining patterns were consistent in non-tesseral and tesseral tissues. Tightly wound collagen fibers were marked by Trichrome's acid fuchsin, but only in the larger and smaller cap zones of little skate polygonal and trabecular tesserae, respectively (*Figure 4A–C*). Also, the presence of elongate lacunae distinguished the cap zone from other tesseral regions (*Figure 4A–C*).

Enough histological features were also visible in SR micro-CT renderings (*Figure 4D–F*), making it possible to segment and analyze the spatial correlation between mineralization patterns and histological features, such as cap zone and spokes (red and blue, respectively, in *Figure 4G–I*). Segmentation of 3D renderings of deep surface views showed that hypermineralized spokes were localized to non-tesseral and tesseral trabecular mineralization patterns in ratfish and little skate, respectively (blue in *Figure 4J–L*). Unlike the well-organized spokes seen in trabecular patterns of little skate tesserae, spokes in ratfish trabecular mineralized tissues were patchy and less organized. Therefore, both non-tesseral and tesseral trabecular mineralization patterns correlated with, and appeared to derive from, spokes and non-spoke regions within body zones.

Unlike little skate polygonal tesserae, trabecular mineralized tissues in ratfish and trabecular tesserae in skate both displayed identical trabecular mineralization patterns whether viewed from the superficial or deep surfaces of the cartilage (*Figure 4J, K, M and N*). The smaller cap zones in trabecular tesserae were discernible on the superficial surface, but they were not as laterally extensive as cap zones of polygonal tesserae (*Figure 4H, I, N and O*). Thus, larger and laterally extensive cap zones appeared to be the reason why polygonal tesserae displayed a superficial polygonal mineralization pattern.

## Ratfish centra contained areolar mineralized tissue

To clarify whether ratfish, catshark, and little skate have areolar mineralized tissue, morphological and histological features of centra were revealed by desktop micro-CT and section histology. Precaudal vertebrae in ratfish had multiple rings of centra per vertebral segment (i.e. unit with paired neural arches and basidorsals; *Figure 5A*). Each centrum in ratfish precaudal vertebrae demonstrated a compact mineralization pattern like that of centra in the catshark and little skate (*Figure 5A–C*). However, no ratfish centrum displayed the typical biconcave gross morphology associated with elasmobranch centra (*Figure 5A–C*; *Atake et al., 2019*; *Clement, 1992*; *Criswell et al., 2017*).

Alizarin red and Alcian blue section histology showed that centra in ratfish, catshark, and little skate comprised compact mineralized tissue spanning different numbers of centrum layers. The ratfish centrum only had mineralized tissue in the middle layer, but an outer layer of the catshark centrum had mineralized cartilage, while inner and outer layers of the little skate centrum contained mineralized cartilage (*Figure 5D–I*). Areolar mineralized tissues in the middle layer of the centrum of ratfish, catshark, and little skate had elongate cell lacunae organized in concentric lamellae (inserts in *Figure 5G–I*; *Atake et al., 2019*; *Criswell et al., 2017*; *Eames et al., 2007*), and they were birefringent (data not shown). Markers of cartilage differed in areolar mineralized tissue among ratfish,

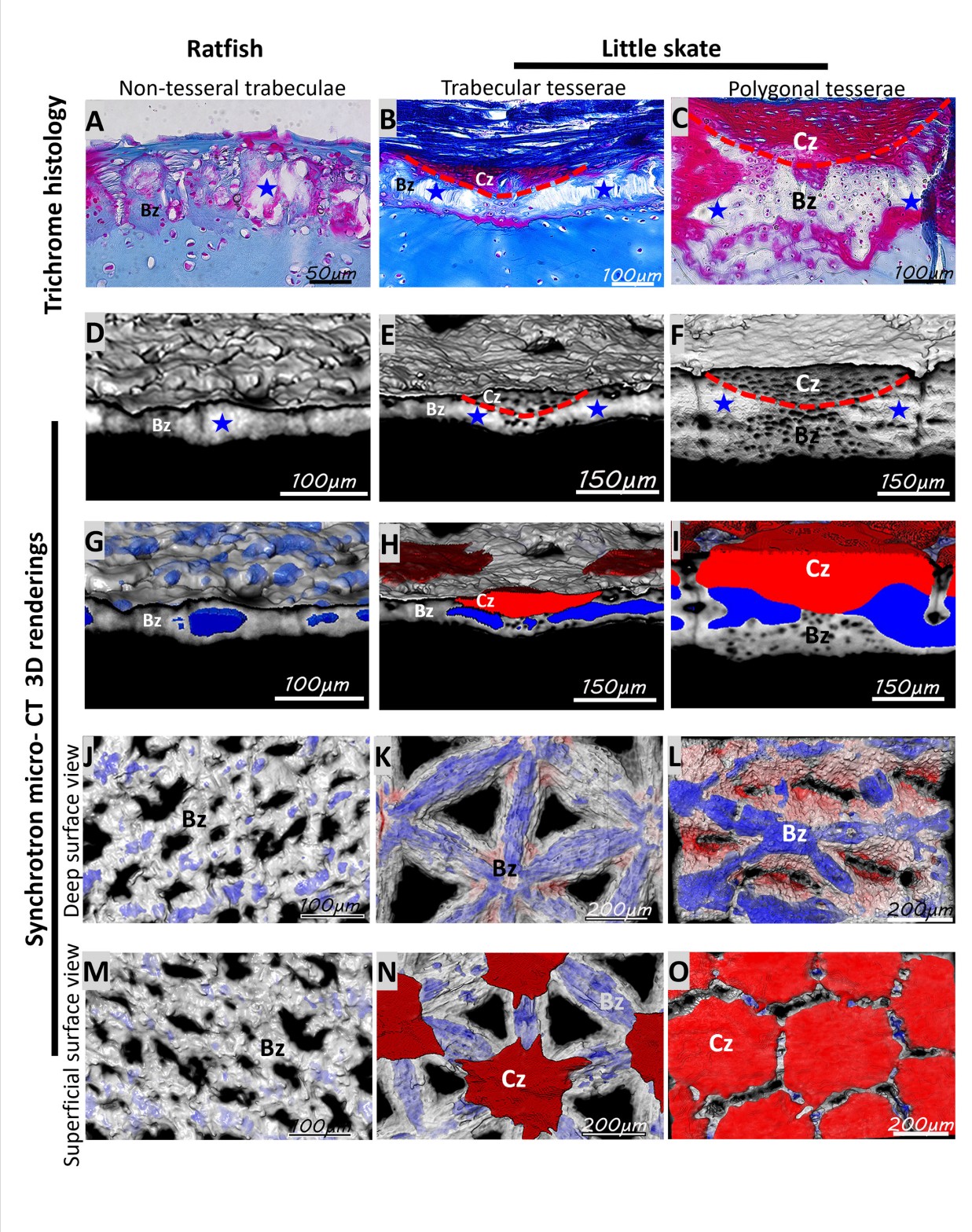

**Figure 4.** Micro-CT segmentation clearly correlated histological features and mineralization patterns. (**A–C**) Trichrome staining showed that spokes in the body zone of ratfish non-tesseral trabeculae (**A**) or little skate trabecular (**B**) or polygonal (**C**) tesserae had low cellularity and were unstained, while intense acid fuchsin staining marked the smaller and larger cap zones of little skate trabecular (**B**) and polygonal (**C**) tesserae, respectively. (**D–I**) 3D renderings of synchrotron micro-CT data showed that hypermineralized spokes (*, segmented in blue) were in the body zone of ratfish non-tesseral trabeculae (**D, G**) or little skate trabecular (**E, H**) or polygonal (**F, I**) tesserae. Tesseral cap zones could also be segmented clearly (red) in little skate (**E,**

*Figure 4 continued on next page*

*Figure 4 continued*

**F, H, I**). (**J–O**) Segmented 3D renderings demonstrated that spokes in ratfish were patchy in deep (**J**) or superficial (**M**) views, while little skate spokes displayed a radiating pattern traversing most of the trabecular patterns in either trabecular (**K, N**) or polygonal (**L, O**) tesserae. The smaller cap zone in trabecular tesserae (**N**) was not laterally extensive enough to display a superficial polygonal pattern like that of polygonal tesserae (**O**). Bz, body zone; Cz, cap zone. Scale bars: as indicated.

catshark, and little skate (*Figure 5J–O*). In ratfish, areolar mineralized tissue had relatively strong Safranin O staining and Col2 immunofluorescence (*Figure 5J and M*). Areolar mineralized tissue in catshark demonstrated weak Safranin O staining and Col2 immunofluorescence (*Figure 5K and N*). By contrast, areolar mineralized tissue in little skate did not display Safranin O staining or Col2 immunofluorescence (*Figure 5L and O*). Overall, these data suggested that ratfish have areolar mineralized tissue.

## Qualitative and quantitative features of centra in the adult ratfish were similar to those of little skate embryos

The biconcave centra morphology is thought to be acquired in elasmobranchs during development when the perichordal sheath becomes constricted by mesenchymal cells (*Arratia et al., 2001*; *Gadow and Abbott, 1895*; *Kölliker, 1860*). To shed light on differences in adult centra morphology, morphological and histological features of centra in a developmental series of little skate were compared to those of adult ratfish. In little skate embryos and hatched juveniles, desktop micro-CT renderings showed mineralization in the centrum and pairs of neural arches, basidorsals, and hemal arches of each vertebral segment (*Figure 6A–C*). Expanding upon recent work showing that anterior and posterior segments of somites fuse during skate vertebral development, similar to tetrapods (*Criswell and Gillis, 2020*), independent anterior and posterior mineralizations of each interneural appeared to later fuse during development. For example, independent mineralization of the posterior and anterior interneurals was visible in stage 32 little skate embryos, whereas the dorsal and ventral aspects of each appeared fused by stage 33, and they were fully fused in a 6.5-cm disc width (DW) juvenile (*Figure 6A–C*). In adult ratfish, each vertebral segment of paired neural arches and basidorsals had an average of six centra (*Figure 6D*). Digital segmentation of SR micro-CT renderings showed that centra in little skate stage 32 embryos were slightly constricted in the middle of each element's anterior-posterior axis, and the extent of this constriction progressed through 6.5-cm DW juveniles, when the classic biconcave morphology was present (*Figure 6E–G*). SR micro-CT renderings showed that adult ratfish centra were constricted to a similar extent as younger centra in little skate embryos (*Figure 6E, F and H*). In little skate embryos, Alcian blue and Alizarin red section histology showed that the areolar mineralized tissue initiated in the middle layer of the centrum and had elongate cell lacunae organized in concentric lamellae, similar to that observed in adult ratfish (*Figure 5D*; *Figure 6I and J*). At later developmental stages of little skate, such as 6.5-cm DW juveniles, the outer layer of the centrum also demonstrated mineralized cartilage (*Figure 6K*).

To further confirm these morphological and histological similarities of centra in little skate embryos and adult ratfish, TMDs of centra in the developmental series of little skate were compared to that of adult ratfish. TMD of little skate centra increased significantly from stage 32 to stage 33 embryos (p=9.5 × 10$^{-3}$), while TMD of 6.5-cm DW juveniles was only significantly higher than TMD of stage 32 embryos (*Figure 6L*; p=9.6 × 10$^{-4}$). TMD of adult ratfish centra was significantly higher than TMD of centra in little skate stage 32 embryos, but it was statistically indistinguishable from TMD of centra in little skate stage 33 embryos and 6.5-cm DW juveniles (*Figure 6L*). Therefore, adult ratfish centra had morphological, histological, and TMD similarities with centra from little skate embryos, particularly those at stage 33.

## Discussion

Which tissues are common in the extant chondrichthyan endoskeleton? Bone-like and areolar tissues have been characterized in several members of the chondrichthyan subclass Elasmobranchii, such as sharks, skates, and rays (*Atake et al., 2019*; *Berio et al., 2021*; *Bordat, 1987*; *Kemp and Westrin, 1979*; *Ørvig, 1951*; *Peignoux-Deville et al., 1982*; *Seidel et al., 2016*). In addition, polygonal tesserae are traditionally associated with chondrichthyans (*Atake et al., 2019*; *Kemp and Westrin,*

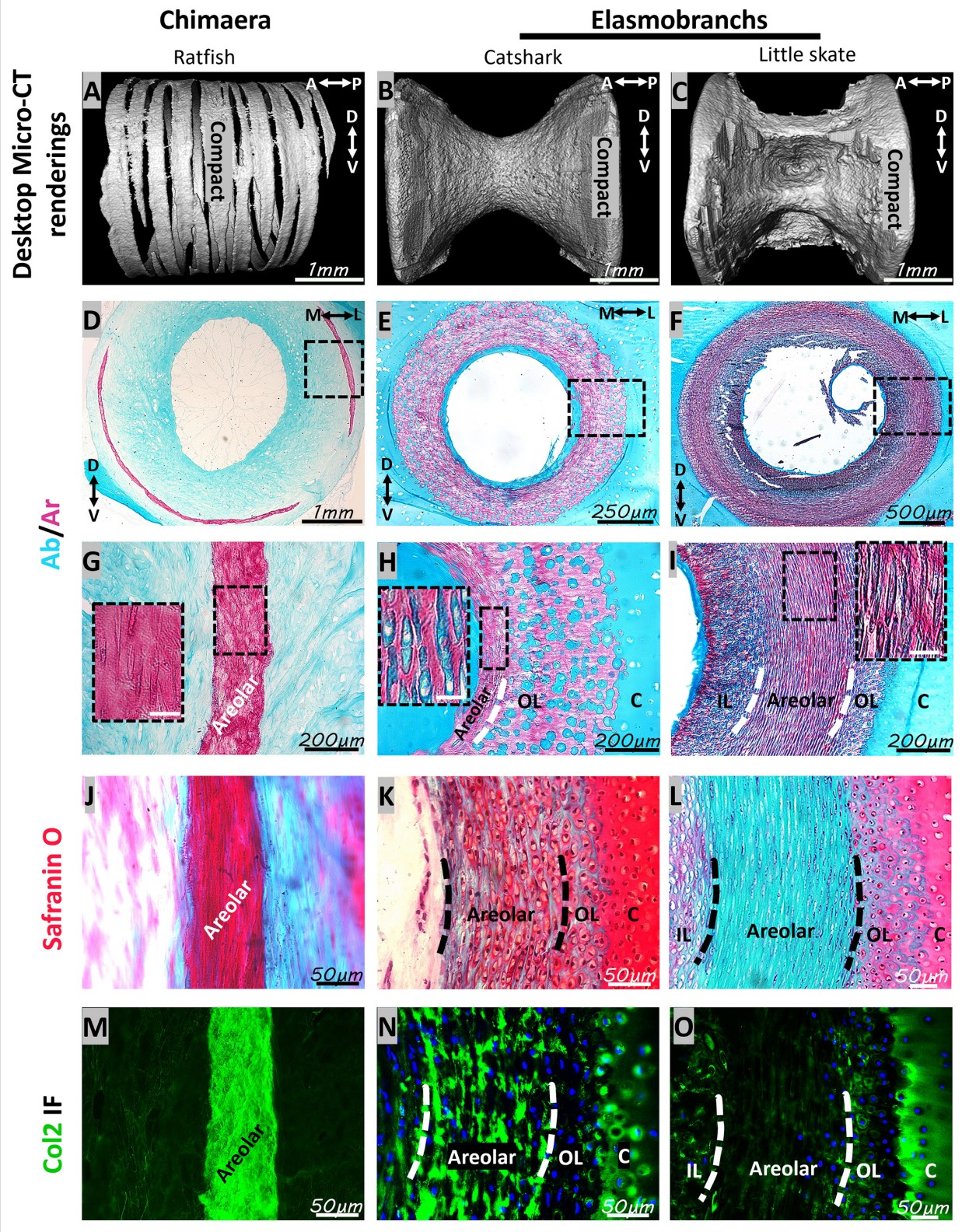

**Figure 5.** Ratfish centra had the same features as areolar mineralized tissues in little skate and catshark. (**A–C**) Micro-CT renderings demonstrated that centra in ratfish (**A**), catshark (**B**), and little skate (**C**) all had a compact mineralization, but ratfish centra were organized in multiple rings and did not display the gross biconcave morphology seen in catshark or little skate. (**D–O**) Alcian blue/Alizarin red and Safranin O section histology and Col2 immunostaining demonstrated that mineralized centra in ratfish (**D, G, J**), catshark (**E, H, K**), and little skate (**F, I, L**) all comprised a middle

*Figure 5 continued on next page*

*Figure 5 continued*

layer with elongate lacunae in concentric lamellae, but in ratfish this was mineralized cartilage, while catshark and little skate had little to no Safranin O or Col2 staining, respectively, in this region. Catshark mineralized centra also comprised the outer layer, while little skate mineralized centra also comprised both inner and outer layers. These additional centra layers had rounded cells embedded in a matrix of weak Safranin O staining and Col2 immunofluorescence. Regions for hi-mag views in panels G–O are indicated in corresponding panels of D–F; regions for hi-mag inserts in panels G–I are indicated in those panels. A, anterior; Ab, Alcian blue; Ar, Alizarin red; c, cartilage; Col2, collagen type 2; D, dorsal; IF, immunofluorescence; IL, inner layer; L, lateral; M, medial; OL, outer layer; P, posterior; V, ventral. Scale bars: as indicated; inserts in G–I are 20 µm.

*1979*; *Maisey, 2013*; *Marramà et al., 2019*; *Seidel et al., 2016*; *Wilga and Ferry, 2015*), but trabecular tesserae were recently characterized in skates and rays (*Atake et al., 2019*; *Jayasankar et al., 2020*).

Two main qualities of this paper go a long way in revealing features of the ancestral chondrichthyan endoskeleton. First, we explicitly identified discrete character states of extant chondrichthyan endoskeletons (*Table 1*), helping to unify terminology of important features for the growing chondrichthyan research community. Second, we presented extensive, side-by-side comparative data across various extant chondrichthyans, two elasmobranch species (small-spotted catshark and little skate), and one chimaera/holocephalan (spotted ratfish). Inclusion of ratfish is another critical aspect of this paper, because holocephalans are generally understudied but must be considered when inferring ancestral chondrichthyan traits. Importantly, previous characterization of skeletal mineralization in other adult chimaeridae neither focused on segmented neural arches to analyze bone-like tissues or areolar mineralization, nor clarified histological zones of tesserae (*Berio et al., 2021*; *Debiais-Thibaud, 2018*; *Pears et al., 2020*; *Seidel et al., 2020*).

Our new data and other published data across various extant and fossil chondrichthyan species can be compiled to compare character states of the extant endoskeleton and infer the ancestral condition (*Figure 7*). In what appears to be a novel character trait of crown chondrichthyans, trabecular mineralization occurs in many endoskeletal regions in several elasmobranch and chimaera species, including some fossil chondrichthyans (*Atake et al., 2019*; *Coates et al., 2018*; *Jayasankar et al., 2020*; *Maisey et al., 2021*; *Ørvig, 1951*). Among extant chondrichthyans, trabecular mineralization always forms near the surface of cartilage, but the exact mineralization pattern varies. Little skate, along with all other batoids and some selachians examined, has arrayed patterns of discrete trabecular units, similar to the repeating polygonal units of traditional tesserae, supporting their description as trabecular tesserae. On the other hand, a non-arrayed trabecular mineralization pattern, not emphasized in chondrichthyans before, appears to be present in catshark and some other selachians and in ratfish and all other chimaeras examined, including *Callorhinchus milii* and *Chimaera monstrosa* (*Pears et al., 2020*; *Seidel et al., 2020*). To distinguish this pattern from the regular, arrayed patterns of polygonal and trabecular tesserae, we propose the term 'non-tesseral trabecular mineralization'. Given that some fossil chondrichthyans, such as *Gladbachus*, *Climatius*, *Doliodus,* and *Cobelodus*, also appear to display this mineralization pattern in their endoskeletons (*Coates et al., 2018*; *Maisey et al., 2021*; *Ørvig, 1951*), non-tesseral trabecular mineralization might be a new plesiomorphic (ancestral) character of chondrichthyans (*Maisey et al., 2021*).

Bone-like tissues do not appear to be shared among extant chondrichthyans, while areolar mineralization does seem to be shared (*Figure 7*). Morphological and histological features of bone-like tissues in neural arches or tesseral cap zones are present in catshark, little skate, and many other elasmobranch species, but absent in ratfish and *C. monstrosa* (*Atake et al., 2019*; *Berio et al., 2021*; *Eames et al., 2007*; *Kemp and Westrin, 1979*; *Seidel et al., 2020*; *Seidel et al., 2017*; *Seidel et al., 2016*; *Wurmbach, 1932*). While these data refute the hypothesis that bone-like tissues generally are a symplesiomorphic (shared ancestral) feature of extant chondrichthyans, they further support the hypothesis that neural arch perichondral bone-like tissue is a synapomorphic (shared derived) character of elasmobranchs, but subsequently lost in several lineages (*Atake et al., 2019*; *Berio et al., 2021*; *Maisey et al., 2021*). On the other hand, similar features of centra are observed in ratfish, catshark, little skate, and many other extant chondrichthyans examined (*Atake et al., 2019*; *Criswell et al., 2017*; *Dean and Summers, 2006*; *Debiais-Thibaud, 2019*; *Eames et al., 2007*; *Porter et al., 2006*). Since most data on areolar mineralization focused on elasmobranchs (*Criswell et al., 2017*; *Eames et al., 2007*; *Ridewood and MacBride, 1921*), histological features of multiple-ringed centra in other chimaera should be determined (*Didier, 1995*). Stem chondrichthyans did not appear to

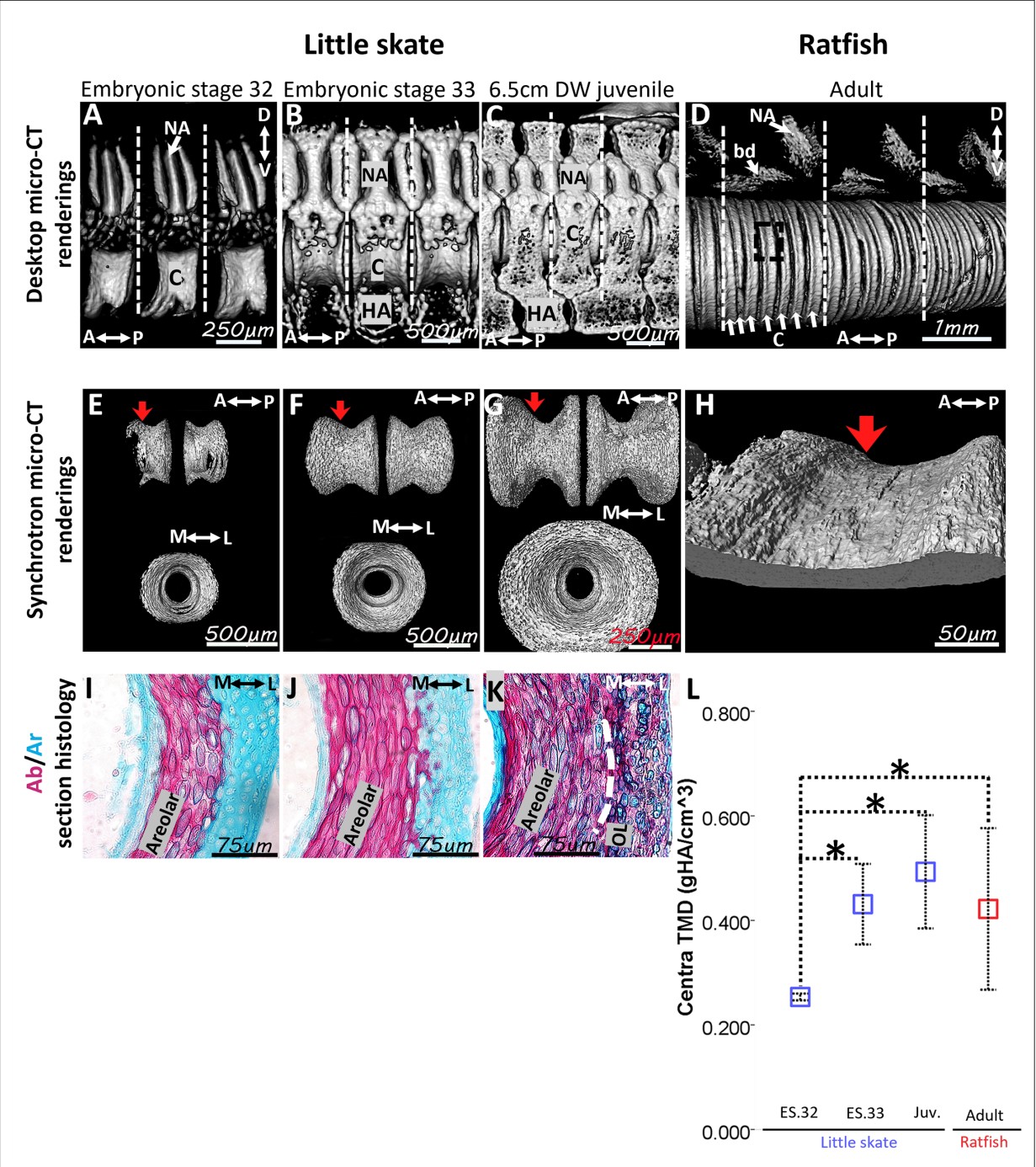

**Figure 6.** Morphological and histological features of centra were shared between little skate embryos and juveniles and adult ratfish. (**A–D**) Desktop micro-CT renderings showed centra and other mineralized structures in vertebral segments (dashed lines) of little skate embryos and juveniles (**A–C**) and adult ratfish (**D**). (**E–H**) Synchrotron micro-CT renderings of centra in little skate embryos and juveniles (**E–G**) and adult ratfish (**H**) showed that centra in the ratfish were slightly constricted, like those of little skate embryos (red arrows). (**I–K**) Alcian blue/Alizarin red section histology showed that areolar mineralized tissue constituted the only centra layer in little skate embryos (**I, J**), whereas an outer mineralized centra layer was present in 6.5-cm disc width (DW) juveniles (**K**). (**L**) Tissue mineral density (TMD) of centra in ratfish was similar to those of stage 33 embryos and 6.5-cm DW juveniles of little skate. A, anterior; Ab, Alcian blue; Ar, Alizarin red; bd, basidorsal; C, centrum; HA, hemal arch; L, lateral; M, medial; NA, neural arch; OL, outer layer; P, posterior.* indicates statistically significant comparison (p<0.05). Scale bars: as indicated.

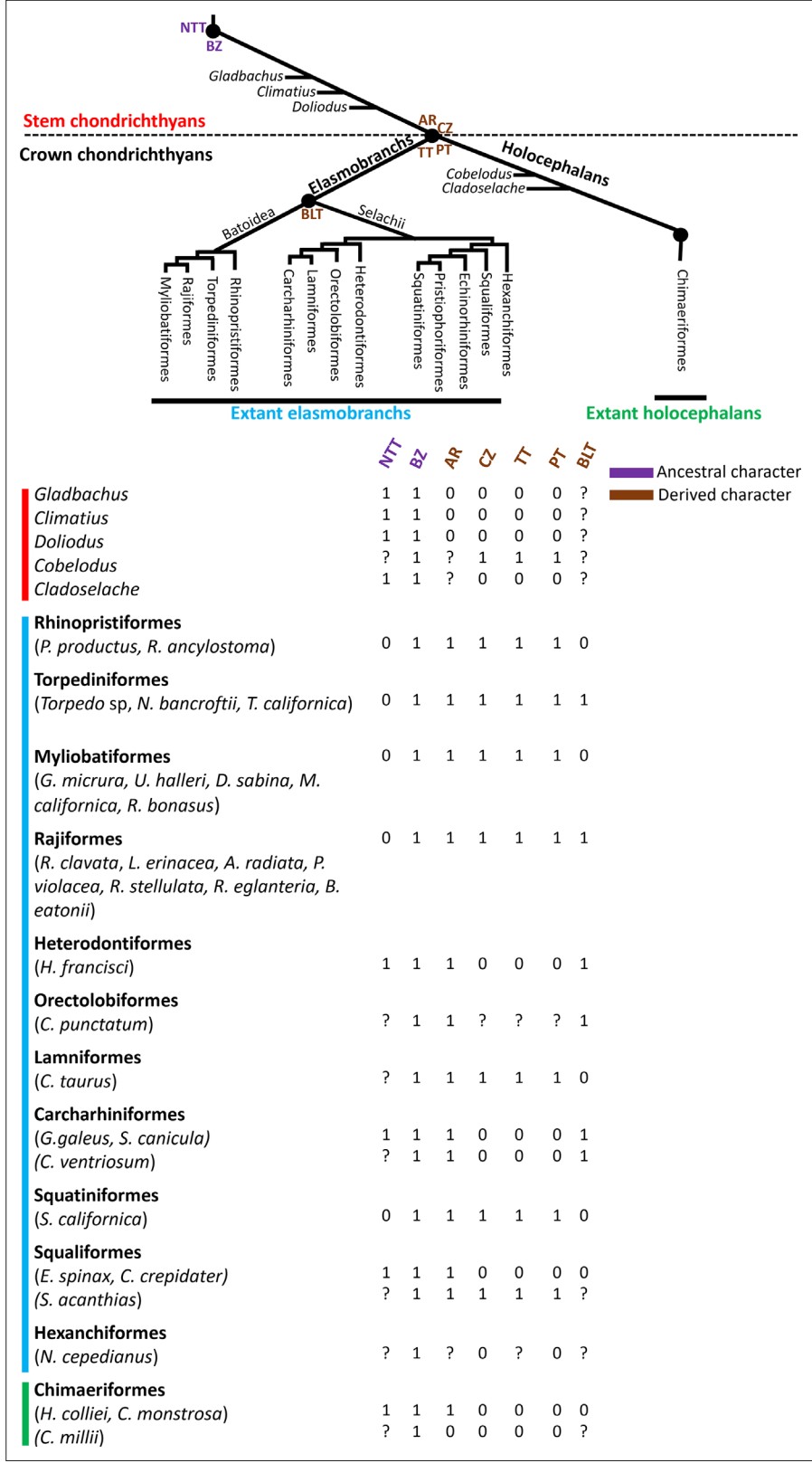

**Figure 7.** Character states of mineralized tissues in various extant and fossil chondrichthyan species suggest shared and derived features. Non-tesseral trabecular mineralization (NTT) is likely a symplesiomorphic (shared ancestral) character of stem and crown chondrichthyans, because it is pervasive in fossil and extant chondrichthyans. Polygonal (PT) and trabecular (TT) tesseral mineralization patterns appear to be absent in

*Figure 7 continued on next page*

*Figure 7 continued*

extant holocephalans and stem chondrichthyans, but their presence in some extant elasmobranchs and a fossil holocephalan suggests that these tesseral mineralizations are synapomorphic (shared derived) characters of crown chondrichthyans, subsequently lost in extant holocephalans. Areolar (AR) mineralized tissue is present in extant elasmobranchs and holocephalans, but its absence in stem chondrichthyans suggests that it is a synapomorphic character of crown chondrichthyans. The absence of neural arch bone-like tissue (BLT) in holocephalans further supports the conclusion that it is a synapomorphic character of elasmobranchs. The presence of a histological body zone (BZ) in either tesseral or non-tesseral mineralized tissues of stem and crown chondrichthyans suggests that the body zone is a symplesiomorphic character of all chondrichthyans. The histological cap zone (CZ) in some elasmobranchs, along with the inference that polygonal tesserae in a fossil holocephalan means that they had cap zones, suggests that the cap zone is also a synapomorphic character of crown chondrichthyans. 0, 1, and ? indicate absence, presence, and unknown, respectively, of characters for the listed species. See (*Supplementary file 2*) for downloadable character matrix.

mineralize their centra, so data from extant chondrichthyans argue that areolar mineralized tissue is likely a synapomorphic character of crown chondrichthyans (*Frey et al., 2019*; *Maisey et al., 2021*; *Miles, 1970*).

Cap and body zones are histological terms that traditionally describe chondrichthyan tesserae (*Dean and Summers, 2006*; *Kemp and Westrin, 1979*), but data from adult samples of ratfish, catshark, little skate, and several other species argue that the body zone is the only histological zone of endoskeletal mineralization shared by extant chondrichthyans (*Figure 7*). Recent histological data of tesserae are few and limited to sharks, skates, and rays, where cap and body zones have been reported (*Atake et al., 2019*; *Berio et al., 2021*; *Eames et al., 2007*; *Enault et al., 2015*; *Kemp and Westrin, 1979*; *Seidel et al., 2017*). While birefringent fibers marked the cap zones in both polygonal and trabecular tesserae in little skate, non-tesseral trabeculae in catshark and ratfish did not have discernible cap zones. In contrast to commonly accepted knowledge, these catshark data suggest that even sharks do not always have the cap zone. Taken alongside recent published data (*Maisey et al., 2021*; *Pears et al., 2020*; *Seidel et al., 2020*), no living species of chimaera likely has cap zones. On the other hand, Safranin O staining and Col2 immunofluorescence showed that histological body zones generated the mineralized endoskeletal cartilage of ratfish, catshark, and little skate. In total, these data suggest that the ancestral stem chondrichthyan generated most of its endoskeletal mineralization with histological body zones, although as the next paragraph highlights, cap zones might have been lost secondarily in chimaera and some shark species.

Digitally segmented 3D regions of micro-CT renderings showed, for the first time, exact spatial correlations between histological zones and mineralization patterns, potentially impacting histological interpretations of fossils. Strikingly, large and laterally extensive cap zones were the basis of the traditional polygonal tesseral mineralization pattern in little skate. Accordingly, similar polygonal tesserae recently described in fossil holocephalans likely have cap zones in addition to body zones and a trabecular tesseral mineralization underlying the polygonal mineralization pattern, and these might have been lost secondarily during holocephalan evolution (*Figure 7*; *Atake et al., 2019*; *Pears et al., 2020*). Increased descriptive power from synchrotron imaging of polygonal tesserae in fossil elasmobranchs and holocephalans can clarify this point. Two other correlations were identified here in body zones. First, spokes were located in body zones. Second, spoke and non-spoke regions of the body zone comprised the trabecular mineralization patterns, regardless of whether these were organized as arrayed tesserae or not. Spokes appeared much more spatially organized in polygonal tesserae and trabecular tesserae than in the non-tesseral trabecular mineralization pattern. By extension of the body zone correlation with trabecular mineralization, trabecular mineralization of some fossil elasmobranchs and holocephalans like *Palaeobates polaris* and *Cladoselache* might derive solely from body zones (*Maisey et al., 2021*; *Ørvig, 1951*). Given such dramatic variation in extant and fossil chondrichthyans, cellular and molecular mechanisms underlying the evolution of tesseral and non-tesseral mineralization patterns are a wide open field of future study.

Finally, three independent features suggested that the adult ratfish endoskeleton shared features of the embryonic little skate endoskeleton. First, TMD was significantly lower in a number of adult ratfish endoskeletal elements, compared to adult catshark and little skate. In fact, the TMD of adult ratfish centra was indistinguishable from the TMD of embryonic little skate centra. Second, in contrast to previous assertions that mineralized centra in chimaeras were unconstricted (*Didier, 1995*; *Gadow*

*and Abbott, 1895*), micro-CT renderings of ratfish centra showed slight constriction, which was like those of centra in little skate embryos. Third, even histological similarities of centra in little skate embryos and adult ratfish were discovered. Both had elongate cell lacunae organized in concentric lamellae only in the mineralized middle layer of the centrum.

A tantalizing possibility is that the similarities between adult ratfish and embryonic little skate endoskeletons represent the effects of a phenomenon termed paedomorphism, or the derived condition of an adult species retaining embryonic traits of an ancestor (*Garstang, 1922*; *Liem et al., 2001*). Given lower TMD in many adult ratfish endoskeletal elements, the ratfish might generally have paedomorphic endoskeletal development, because as mineralized skeletal tissues develop, their TMD tends to increase (*Forbes, 1976*). The genetic basis for reduced skeletal mineralization in chimaeras should be investigated further, especially since the only published full chondrichthyan genome was from *Callorhinchus milii* (Callorhinchidae; *Venkatesh et al., 2014*), and Callorhinchidae is the only chimaera family, for example, that does not mineralize their centra (*Figure 7*; *Didier, 1995*; *Gadow and Abbott, 1895*). Our findings on slight constriction of centra in both adult ratfish and embryonic little skate independently supported a previous assertion that the perichordal sheath in adult chimaeras remained at a stage corresponding to that of embryonic elasmobranchs (*Gadow and Abbott, 1895*). Additional future discoveries, however, would be needed to support the idea that the endoskeleton of ratfish (perhaps even extant holocephalans generally) is paedomorphic with respect to that of the common ancestor of holocephalans and elasmobranchs. First, these traits (lower endoskeletal TMD, slight morphological constriction of centra, and elongated lacunae in the only mineralized [middle] layer of centra) must be present in this last common ancestor. Second, primitive holocephalans must have altered these traits. Hopefully, paleontologists can resolve these issues.

In summary, analyses of extant and fossil data suggest different features of the chondrichthyan endoskeleton appeared (and disappeared) at different times during evolution of this fascinating clade (*Figure 7*). The widespread distribution of non-tesseral trabecular mineralization among fossil stem and extant crown chondrichthyans argues that this is a symplesiomorphic chondrichthyan character. Areolar mineralization is widespread among extant chondrichthyan species, but absent in early stem chondrichthyans, suggesting it evolved in a later stem chondrichthyan that gave rise to holocephalans and elasmobranchs. The presence of tesserae in extant elasmobranchs and the fossil holocephalan *Cobelodus* argues that this trait was lost secondarily in extant holocephalans. Bone-like tissues in cap zones or neural arches appear restricted to extant elasmobranchs. Extant species have correlations between the histological body zone and any form of trabecular mineralization on the one hand, and the cap zone and polygonal tesserae on the other. Given these correlations, the body zone might also be symplesiomorphic for all chondrichthyans, while the cap zone might be synapomorphic for crown chondrichthyans, secondarily lost in extant holocephalans.

## Materials and methods

### Specimens

All samples were approved for use under the University of Saskatchewan ethical protocol (AUP 20130092). Vertebrae (precaudal and caudal) from three samples of adult small-spotted catshark with total lengths (TL) of 31 cm were provided by the University of Montpellier Aquarium (Planet Ocean Montpellier). Adult, juvenile, and embryonic samples of the little skate were obtained from Marine Biological Labs (Falmouth, MA, USA). Four samples of adult little skate with TL ranging from 43.5 cm to 47.5 cm were sampled (*Supplementary file 1*). Little skate juveniles (n=7) were staged by measuring their TL and DW and ranged from 5.5 cm DW, 10 cm TL to 6.5 cm DW, 11 cm TL (*Supplementary file 1*; *Stehmann, 2002*). The average DW and TL of stage 32 embryos (n=6) and stage 33 embryos (n=6) were 3.2 cm DW, 7.5 cm TL and 3.5 cm DW, 8 cm TL, respectively (*Supplementary file 1*; *Maxwell et al., 2008*; *Vazquez et al., 2020*). Five samples of adult spotted ratfish with TL ranging from 28.5 cm to 45 cm were captured by deep-water trawl in the San Juan Islands, WA, frozen within 1 or 2 hr of catch, and kept frozen for several years until thawed for experiments (*Supplementary file 1*).

Regions of interest, such as ceratohyal, synarcual, and precaudal and caudal vertebrae, were dissected from the little skate, catshark, and ratfish samples (*Supplementary file 1*). Dissected tissues were preserved by fixation in 4% paraformaldehyde in PBS (pH 7.4) and dehydrated through a graded ethanol series.

## Micro-CT imaging and data processing

Desktop micro-CT imaging of ratfish ROIs, little skate ROIs, and catshark precaudal vertebrae was done using SkyScan 1172 desktop microtomograph (Bruker SkyScan, Kontich, Belgium). Desktop micro-CT projections were acquired using a 0.5 mm aluminium filter at 40 kV and 250 μA, 100 ms exposure time, and 10 μm voxel resolution, and reconstructed using NRecon (Bruker SkyScan, Kontich, Belgium). A lower resolution projection (i.e. 26 μm voxel, 700 ms exposure time) of an anterior region of the ratfish was acquired in parts (*Figure 1A*). Scans from the different parts were later stitched together using Dragonfly v2021.1 (Object Research Systems, Canada). Desktop imaging of catshark caudal vertebrae was performed using CT scanner: EasyTom 150 scanner at 70 kV, 80 μA, 1.43 s exposure time, and 6 μm voxel resolution. Reconstructions were done with Xact software (v11025), and images were rendered with Avizo Lite software (v2019.3). TMD of ratfish, little skate, and catshark ROIs was quantitated on SkyScan 1172 desktop micro-CT images using CTAnalyzer v1.16 (CTAn, Bruker SkyScan, Kontich, Belgium) as previously described (*Atake et al., 2019*).

SR-based micro-CT imaging of ratfish and little skate was done on the Biomedical Imaging and Therapy-Insertion Device 05ID-2 (BMIT-ID) line at the Canadian Light Source. SR micro-CT projections were acquired with a 28 keV photon energy, 1.44 μm voxel resolution, 1 s exposure time, and a sample to detector distance of 9 cm. Reconstruction of SR micro-CT projections and phase retrieval was done using UFO-KIT software (https://github.com/ufo-kit/ufo-core) (*Vogelgesang and Farago, 2025*). A delta/beta ratio of 200 was applied during phase retrieval. 3D rendered volumes and 2D virtual slices of reconstructed data were generated with Amira 6.0 (FEI Group, USA). 3D segmentation and color-coding of morphological features were done using segmentation editor tools in Amira.

## Histological and immunofluorescence assays

Tissues for section histology were not demineralized before sectioning. Tissues were embedded in optimum cutting temperature compound (Tissue Tek, Torrance, CA, United States), and serial tissue sections of 10 μm thickness were made with a Cryostar NX50 cryostat (Fisher Scientific, United States).

Tissue sections were stained as described with Safranin O (*Ferguson et al., 1998*), picrosirius red (*Junqueira et al., 1978*), or Milligan's Trichrome (*Ashique et al., 2022*). Alcian blue/Alizarin red staining was done using a modified acid-free protocol (*Eames et al., 2011*). Briefly, tissue sections were washed with 100 mM Tris pH 7.5/10 mM $MgCl_2$, stained with 0.04% Alcian blue/70% EtOH/10 mM $MgCl_2$ pH 7.5, taken through graded EtOH series (80% EtOH/100 mM Tris pH 7.5/10 mM MgCl2; 50% EtOH/100 mM Tris pH 7.5; 25% EtOH/100 mM Tris pH 7.5), stained with 0.1% Alizarin red/0.1% KOH pH 7.5, de-stained with two washes of 0.1% KOH, and dehydrated through 25% EtOH, 50% EtOH, 80% EtOH, and 100% EtOH series before coverslipping. Tissue sections were demineralized using 5% EDTA for 10 min before staining with Safranin O to improve staining of mineralized cartilage.

To minimize the chances of tissues falling off slides, tissue sections for immunofluorescence were incubated at 55 °C for 15 min before treatments and then washed with PBST (1xPBS/0.5 x triton-X) between subsequent treatments. Tissues were treated separately with trypsin (0.1% in 5% EDTA/1x PBS) and hyaluronidase (0.5% in 1x PBS/0.5 x Triton-X) at 37 °C for 15 min each. The tissues were blocked with 4% goat serum/2% sheep/1xPBS for 1 hr, and then incubated overnight at 4 °C with either Col2 antibody (1:100, II-II6B3, Hybridoma Bank) or blocking solution (negative control). Secondary antibody labeling was done using 488-conjugated goat anti-mouse antibody (1:1000, A32723 Thermo Fisher Scientific) for 3 hr.

## Quantitative and statistical analyses

Statistical analyses of TMDs were performed using SPSS V.22 (SPSS). Shapiro–Wilk test was used to test for normal distribution of data. To assess differences among means, one-way ANOVA was used followed by Tukey HSD or Games-Howell post hoc analyses, depending upon whether homogeneity of variance assumption (Levene's test) was met or not, respectively.

## Acknowledgements

This work was funded by Natural Sciences and Engineering Research Council (NSERC) grants RGPIN 435655–201 and RGPIN 2014–05563 awarded to BFE. The authors are very thankful to Adam Summers of Friday Harbor in the San Juan Islands for providing the spotted ratfish samples and some feedback

on the spotted ratfish data. Research described in this paper was performed at the BMIT facility at the Canadian Light Source, which is supported by Canada Foundation for Innovation, Natural Sciences and Engineering Research Council of Canada, the University of Saskatchewan, Western Economic Diversification Canada, the National Research Council Canada, and the Canadian Institutes of Health Research. We acknowledge the MRI platform member of the national infrastructure France-BioImaging supported by the French National Research Agency (ANR-10-INBS- 04, "Investments for the future"), the labex CEMEB (ANR-10-LABX-0004) and NUMEV (ANR-10-LABX-0020).

## Additional information

### Funding

| Funder | Grant reference number | Author |
| --- | --- | --- |
| Natural Sciences and Engineering Research Council of Canada | RGPIN 435655-201 | B Frank Eames |
| Agence Nationale de la Recherche | ANR-10-INBS- 04 | Melanie Debiais Thibaud |
| Centre Méditerranéen de l'Environnement et de la Biodiversité | ANR-10-LABX-0004 | Melanie Debiais Thibaud |
| LabEx NUMEV | ANR-10-LABX-0020 | Melanie Debiais Thibaud |
| Natural Sciences and Engineering Research Council of Canada | RGPIN 2014-05563 | B Frank Eames |

The funders had no role in study design, data collection and interpretation, or the decision to submit the work for publication.

### Author contributions

Oghenevwogaga Joseph Atake, Conceptualization, Funding acquisition, Project administration, Resources, Supervision, Writing – review and editing, Data curation; Fidji Berio, Conceptualization, Funding acquisition, Resources, Supervision, Writing – review and editing; Melanie Debiais Thibaud, Supervision, Funding acquisition, Project administration, Writing – review and editing; B Frank Eames, Conceptualization, Resources, Supervision, Funding acquisition, Project administration, Writing – review and editing

### Author ORCIDs

Oghenevwogaga Joseph Atake ⓘ https://orcid.org/0000-0003-2803-0183
Fidji Berio ⓘ https://orcid.org/0000-0003-0810-9783
B Frank Eames ⓘ https://orcid.org/0000-0002-8200-3760

### Ethics

All of the animals were handled according to approved institutional animal care and use committee (IACUC) protocols. All samples were approved for use under the University of Saskatchewan ethical protocol (AUP 20130092).

Reviewer #2 (Public review): https://doi.org/10.7554/eLife.94900.4.sa1
Author response https://doi.org/10.7554/eLife.94900.4.sa2

## Additional files

### Supplementary files
MDAR checklist

Supplementary file 1. Sample size, body measurements, and regions of interest analyzed for little

skate, catshark, and ratfish samples.

Supplementary file 2. Survey of mineralized tissues in various extant and fossil chondrichthyan species.

### Data availability

Raw data from microCT projections and quantitative measurements have been deposited at Dryad.

The following dataset was generated:

| Author(s) | Year | Dataset title | Dataset URL | Database and Identifier |
|---|---|---|---|---|
| Atake O, Berio F, Debiais-Thibaud M, Eames BF | 2025 | The holocephalan ratfish endoskeleton shares trabecular and areolar mineralization patterns, but not tesserae, with elasmobranchs little skate and catshark | https://doi.org/10.5061/dryad.0cfxpnwfq | Dryad Digital Repository, 10.5061/dryad.0cfxpnwfq |

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
