## [Editor Report · eLife Assessment]

This **important** study significantly advances our understanding of the skeleton of cartilaginous fishes by using a range of state of the art and complementary approaches to compare the skeleton amongst three cartilagenous fishes (catshark, little skate and ratfish). The evidence presented is **compelling** and likely to impact several fields of study.

---

## [Referee Report · Reviewer #2 (Public review)]

General comment:

This is a very valuable and unique comparative study. An excellent combination of scanning and histological data from three different species is presented. Obtaining the material for such a comparative study is never trivial. The study presents new data and thus provides the basis for an in-depth discussion about chondrichthyan mineralised skeletal tissues.

Comments on previous revisions:

The manuscript has been revised and improved and can be published. A very nice manuscript, indeed. My only recommendation (point of discussion, not a requirement) would still be to think about the claim of paedomorphosis in a holocephalan.

Within the chondrichthyes, how distant holocephali are in relation to elasmobranchii remains uncertain, holocephali are quite a specialised group. Holocephali are also older than Batoidea and Selachii. As paedomorphosis is a derived character, I imagine it is difficult to establish that development in an extant holocephalan is derived compared to development in elasmobranchii. If this type of development would have been typical for the "older" holocephali it would not be paedomorphic. Also, the uncertainty how distant holocephali are from elasmobranchii makes it difficult to identify paedomorphosis with reference to chondrichthyes.

[Editors note: the authors have made further revisions in response to the previous reviews.]

---

## [Author Response]

The following is the authors’ response to the previous reviews

**Reviewer #1 (Public Review):**
Summary:It seems as if the main point of the paper is about the new data related to rat fish although your title is describing it as extant cartilaginous fishes and you bounce around between the little skate and ratfish. So here's an opportunity for you to adjust the title to emphasize ratfish is given the fact that leader you describe how this is your significant new data contribution. Either way, the organization of the paper can be adjusted so that the reader can follow along the same order for all sections so that it's very clear for comparative purposes of new data and what they mean. My opinion is that I want to read, for each subheading in the results, about the the ratfish first because this is your most interesting novel data. Then I want to know any confirmation about morphology in little skate. And then I want to know about any gaps you fill with the cat shark. (It is ok if you keep the order of "skate, ratfish, then shark, but I think it undersells the new data).

The main points of the paper are (1) to define terms for chondrichthyan skeletal features in order to unify research questions in the field, and (2) add novel data on how these features might be distributed among chondrichthyan clades. However, we agree with the reviewer that many readers might be more interested in the ratfish data, so we have adjusted the order of presentation to emphasize ratfish throughout the manuscript.

Strengths:The imagery and new data availability for ratfish are valuable and may help to determine new phylogenetically informative characters for understanding the evolution of cartilaginous fishes. You also allude to the fossil record.

Thank you for the nice feedback.

Opportunities:I am concerned about the statement of ratfish paedomorphism because stage 32 and 33 were not statistically significantly different from one another (figure and prior sentences). So, these ratfish TMDs overlap the range of both 32 and 33. I think you need more specimens and stages to state this definitely based on TMD. What else leads you to think these are paedomorphic? Right now they are different, but it's unclear why. You need more outgroups.

Sorry, but we had reported that the TMD of centra from little skate did significantly increase between stage 32 and 33. Supporting our argument that ratfish had features of little skate embryos, TMD of adult ratfish centra was significantly lower than TMD of adult skate centra (Fig1). Also, it was significantly higher than stage 33 skate centra, but it was statistically indistinguishable from that of stage 33 and juvenile stages of skate centra. While we do agree that more samples from these and additional groups would bolster these data, we feel they are sufficiently powered to support our conclusions for this current paper.

Your headings for the results subsection and figures are nice snapshots of your interpretations of the results and I think they would be better repurposed in your abstract, which needs more depth.

We have included more data summarized in results sub-heading in the abstract as suggested (lines 32-37).

Historical literature is more abundant than what you've listed. Your first sentence describes a long fascination and only goes back to 1990. But there are authors that have had this fascination for centuries and so I think you'll benefit from looking back. Especially because several of them have looked into histology and development of these fishes.I agree that in the past 15 years or so a lot more work has been done because it can be done using newer technologies and I don't think your list is exhaustive. You need to expand this list and history which will help with your ultimate comparative analysis without you needed to sample too many new data yourself.

We have added additional recent and older references: Kölliker, 1860; Daniel, 1934; Wurmbach, 1932; Liem, 2001; Arratia et al., 2001.

I'd like to see modifications to figure 7 so that you can add more continuity between the characters, illustrated in figure 7 and the body of the text.

We address a similar comment from this reviewer in more detail below, hoping that any concerns about continuity have been addressed with inclusion of a summary of proposed characters in a new Table 1, re-writing of the Discussion, and modified Fig7 and re-written Fig7 legend.

Generally Holocephalans are the outgroup to elasmobranchs - right now they are presented as sister taxa with no ability to indicate derivation. Why isn't the catshark included in this diagram?

While a little unclear exactly what was requested, we restructured the branches to indicate that holocephalans diverged earlier from the ancestors that led to elasmobranchs. Also in response to this comment, we added catshark (*S. canicula*) and little skate (*L. erinacea*) specifically to the character matrix.

In the last paragraph of the introduction, you say that "the data argue" and I admit, I am confused. Whose data? Is this a prediction or results or summary of other people's work? Either way, could be clarified to emphasize the contribution you are about to present.

Sorry for this lack of clarity, and we have changed the wording in this revision to hopefully avoid this misunderstanding.

**Reviewer #2 (Public Review):**
General comment:This is a very valuable and unique comparative study. An excellent combination of scanning and histological data from three different species is presented. Obtaining the material for such a comparative study is never trivial. The study presents new data and thus provides the basis for an in-depth discussion about chondrichthyan mineralised skeletal tissues.

many thanks for the kind words

I have, however, some comments. Some information is lacking and should be added to the manuscript text. I also suggest changes in the result and the discussion section of the manuscript.Introduction:The reader gets the impression almost no research on chondrichthyan skeletal tissues was done before the 2010 ("last 15 years", L45). I suggest to correct that and to cite also previous studies on chondrichthyan skeletal tissues, this includes studies from before 1900.

We have added additional older references, as detailed above.

Material and Methods:Please complete L473-492: Three different Micro-CT scanners were used for three different species? ScyScan 117 for the skate samples. Catshark different scanner, please provide full details. Chimera Scncrotron Scan? Please provide full details for all scanning protocols.

We clarified exact scanners and settings for each micro-CT experiment in the Methods (lines 476-497).

TMD is established in the same way in all three scanners? Actually not possible. Or, all specimens were scanned with the same scanner to establish TMD? If so please provide the protocol.

Indeed, the same scanner was used for TMD comparisons, and we included exact details on how TMD was established and compared with internal controls in the Methods. (lines 486-488)

Please complete L494 ff: Tissue embedding medium and embedding protocol is missing. Specimens have been decalcified, if yes how? Have specimens been sectioned non-decalcified or decalcified?Please complete L506 ff: Tissue embedding medium and embedding protocol is missing. Description of controls are missing.

Methods were updated to include these details (lines 500-503).

Results:L147: It is valuable and interesting to compare the degree of mineralisation in individuals from the three different species. It appears, however, not possible to provide numerical data for Tissue Mineral Density (TMD). First requirement, all specimens must be scanned with the same scanner and the same calibration values. This in not stated in the M&M section. But even if this was the case, all specimens derive from different sample locations and have, been preserved differently. Type of fixation, extension of fixation time in formalin, frozen, unfrozen, conditions of sample storage, age of the samples, and many more parameters, all influence TMD values. Likewise the relative age of the animals (adult is not the same as adult) influences TMD. One must assume different sampling and storage conditions and different types of progression into adulthood. Thus, the observation of different degrees of mineralisation is very interesting but I suggest not to link this observation to numerical values.

These are very good points, but for the following reasons we feel that they were not sufficiently relevant to our study, so the quantitative data for TMD remain scientifically valid and critical for the field moving forward. Critically, (1) all of the samples used for TMD calculations underwent the same fixation protocols, and (2) most importantly, all samples for TMD were scanned on the same micro-CT scanner using the same calibration phantoms for each scanning session. Finally, while the exact age of each adult was not specified, we note for Fig1 that clear statistically significant differences in TMD were observed among various skeletal elements from ratfish, shark, and skate. Indeed, ratfish TMD was considerably lower than TMD reported for a variety of fishes and tetrapods (summarized in our paper about icefish skeletons, who actually have similar TMD to ratfish: https://doi.org/10.1111/joa.13537).

In , however, we added a caveat to the paper’s Methods (lines 466-469), stating that adult ratfish were frozen within 1 or 2 hours of collection from the wild, staying frozen for several years prior to thawing and immediate fixation.

Parts of the results are mixed with discussion. Sometimes, a result chapter also needs a few references but this result chapter is full of references.

As mentioned above, we reduced background-style writing and citations in each Results section.

Based on different protocols, the staining characteristics of the tissue are analysed. This is very good and provides valuable additional data. The authors should inform the not only about the staining (positive of negative) abut also about the histochemical characters of the staining. L218: "fast green positive" means what? L234: "marked by Trichrome acid fuchsin" means what? And so on, see also L237, L289, L291

We included more details throughout the Results upon each dye’s first description on what is generally reflected by the specific dyes of the staining protocols. (lines 178, 180, 184, 223, 227, and 243-244)

DiscussionPlease completely remove figure 7, please adjust and severely downsize the discussion related to figure 7. It is very interesting and valuable to compare three species from three different groups of elasmobranchs. Results of this comparison also validate an interesting discussion about possible phylogenetic aspects. This is, however, not the basis for claims about the skeletal tissue organisation of all extinct and extant members of the groups to which the three species belong. The discussion refers to "selected representatives" (L364), but how representative are the selected species? Can there be a extant species that represents the entire large group, all sharks, rays or chimeras? Are the three selected species basal representatives with a generalist life style?

These are good points, and yes, we certainly appreciate that the limited sampling in our data might lead to faulty general conclusions about these clades. In fact, we stated this limitation clearly in the Introduction (lines 126-128), and we removed “representative” from this revision. We also replaced general reference to chondrichthyans in the Title by listing the specific species sampled. However, in the Discussion, we also compare our data with previously published additional species evaluated with similar assays, which confirms the trend that we are concluding. We look forward to future papers specifically testing the hypotheses generated by our conclusions in this paper, which serves as a benchmark for identifying shared and derived features of the chondrichthyan endoskeleton.

Please completely remove the discussion about paedomorphosis in chimeras (already in the result section). This discussion is based on a wrong idea about the definition of paedomorphosis. Paedomorphosis can occur in members of the same group. Humans have paedormorphic characters within the primates, Ambystoma mexicanum is paedormorphic within the urodeals. Paedomorphosis does not extend to members of different vertebrate branches. That elasmobranchs have a developmental stage that resembles chimera vertebra mineralisation does not define chimera vertebra centra as paedomorphic. Teleost have a herocercal caudal fin anlage during development, that does not mean the heterocercal fins in sturgeons or elasmobranchs are paedomorphic characters.

We agree with the reviewer that discussion of paedomorphosis should apply to members of the same group. In our paper, we are examining paedomorphosis in a holocephalan, relative to elasmobranch fishes in the same group (Chrondrichthyes), so this is an appropriate application of paedomorphosis. In response to this comment, we clarified that our statement of paedomorphosis in ratfish was made with respect to elasmobranchs (lines 37-39; 418-420).

L432-435: In times of Gadow & Abott (1895) science had completely wrong ideas bout the phylogenic position of chondrichthyans within the gnathostomes. It is curious that Gadow & Abott (1895) are being cited in support of the paedomorphosis claim.

If paedomorphosis is being examined within Chondrichthyes, such as in our paper and in the Gadow and Abbott paper, then it is an appropriate reference, even if Gadow and Abbott (and many others) got the relative position of Chondrichthyes among other vertebrates incorrect.

The SCPP part of the discussion is unrelated to the data obtained by this study. Kawaki & WEISS (2003) describe a gene family (called SCPP) that control Ca-binding extracellular phosphoproteins in enamel, in bone and dentine, in saliva and in milk. It evolved by gene duplication and differentiation. They date it back to a first enamel matrix protein in conodonts (Reif 2006). Conodonts, a group of enigmatic invertebrates have mineralised structures but these structure are neither bone nor mineralised cartilage. Cat fish (6 % of all vertebrate species) on the other hand, have bone but do not have SCPP genes (Lui et al. 206). Other calcium binding proteins, such as osteocalcin, were initially believed to be required for mineralisation. It turned out that osteocalcin is rather a mineralisation inhibitor, at best it regulates the arrangement collagen fiber bundles. The osteocalcin -/- mouse has fully mineralised bone. As the function of the SCPP gene product for bone formation is unknown, there is no need to discuss SCPP genes. It would perhaps be better to finish the manuscript with summery that focuses on the subject and the methodology of this nice study.

We completely agree with the reviewer that many papers claim to associate the functions of SCPP genes with bone formation, or even mineralization generally. The Science paper with the elephant shark genome made it very popular to associate SCPP genes with bone formation, but we feel that this was a false comparison (for many reasons)! In response to the reviewer’s comments, however, we removed the SCPP discussion points, moving the previous general sentence about the genetic basis for reduced skeletal mineralization to the end of the previous paragraph (lines 435-439). We also added another brief Discussion paragraph afterwards, ending as suggested with a summary of our proposed shared and derived chondrichthyan endoskeletal traits (lines 440-453).

**Reviewer #1 (Recommendations For The Authors):**
Further Strengths and Opportunities:Your headings for the results subsection and figures are nice snapshots of your interpretations of the results and I think they would be better repurposed in your abstract, which needs more depth. It's a little unusual to try and state an interpretation of results as the heading title in a results section and the figures so it feels out of place. You could also use the headings as the last statement of each section, after you've presented the results. In order I would change these results subheadings to:Tissue Mineral Density (TMD)Tissue Properties of Neural ArchesTrabecular mineralizationCap zone and Body zone Mineralization PatternsAreolar mineralizationDevelopmental Variation

Sorry, but we feel that summary Results sub-headings are the best way to effectively communicate to readers the story that the data tell, and this style has been consistently used in our previous publications. No changes were made.

You allude to the fossil record and that is great. That said historical literature is more abundant than what you've listed. Your first sentence describes a long fascination and only goes back to 1990. But there are authors that have had this fascination for centuries and so I think you'll benefit from looking back. Especially because several of them have looked into histology of these fishes. You even have one sentence citing Coates et al. 2018, Frey et al., 2019 and ørvig 1951 to talk about the potential that fossils displayed trabecular mineralization. That feels like you are burying the lead and may have actually been part of the story for where you came up with your hypothesis in the beginning... or the next step in future research. I feel like this is really worth spending some more time on in the intro and/or the discussion.

We’ve added older REFs as pointed out above. Regarding fossil evidence for trabecular mineralization, no, those studies did not lead to our research question. But after we discovered how widespread trabecular mineralization was in extant samples, we consulted these papers, which did not focus on the mineralization patterns per se, but certainly led us to emphasize how those patterns fit in the context of chondrichthyan evolution, which is how we discussed them.

I agree that in the past 15 years or so a lot more work has been done because it can be done using newer technologies. That said there's a lot more work by Mason Dean's lab starting in 2010 that you should take a look at related to tesserae structure... they're looking at additional taxa than what you did as well. It will be valuable for than you to be able to make any sort of phylogenetic inference as part of your discussion and enhance the info your present in figure 7. Go further back in time... For example:de Beer, G. R. 1932. On the skeleton of the hyoid arch in rays and skates. QuarterlyJournal of Microscopical Science. 75: 307-319, pls. 19-21.de Beer, G. R. 1937. The Development of the Vertebrate Skull. The University Press,Oxford.

Indeed, we have read all of Mason’s work, citing 9 of his papers, and where possible, we have incorporated their data on different species into our Discussion and Fig7. Thanks for the de Beer REFs. While they contain histology of developing chondrichthyan elements, they appear to refer principally to gross anatomical features, so were not included in our Intro/Discussion.

Most sections with in the results, read more like a discussion than a presentation of the new data and you jump directly into using an argument of those data too early. Go back in and remove the references or save those paragraphs for the discussion section. Particularly because this journal has you skip the method section until the end, I think it's important to set up this section with a little bit more brevity and conciseness. For instance, in the first section about tissue mineral density, change that subheading to just say tissue mineral density. Then you can go into the presentation of what you see in the ratfish, and then what you see in the little skate, and then that's it. You save the discussion about what other elasmobranch's or mineralizing their neural arches, etc. for another section.

We dramatically reduced background-style writing and citations in each Results section (other than the first section of minor points about general features of the ratfish, compared to catshark and little skate), keeping only a few to briefly remind the general reader of the context of these skeletal features.

I like that your first sentence in the paragraph is describing why you are doing. a particular method and comparison because it shows me (the reader) where you're sampling from. Something else is that maybe as part of the first figure rather than having just each with the graph have a small sketch for little skate and catch shark to show where you sampled from for comparative purposes. That would relate back, then to clarifying other figures as well.

done (also adding a phylogenetic tree).

Second instance is your section on trabecular mineralization. This has so many references in it. It does not read like results at all. It looks like a discussion. However, the trabecular mineralization is one of the most interesting aspect of this paper, and how you are describing it as a unique feature. I really just want a very clear description of what the definition of this trabecular mineralization is going to be.

In addition to adding Table 1 to define each proposed endoskeletal character state, we have changed the structure of this section and hope it better communicates our novel trabecular mineralization results. We also moved the topic of trabecular mineralization to the first detailed Discussion point (lines 347-363) to better emphasize this specific topic.

Carry this reformatting through for all subsections of the results.

As mentioned above, we significantly reduced background-style writing and citations in each Results section.

I'd like to see modifications to figure 7 so that you can add more continuity between the characters, illustrated in figure 7 and the body of the text. I think you can give the characters a number so that you can actually refer to them in each subsection of the results. They can even be numbered sequentially so that they are presented in a standard character matrix format, that future researchers can add directly to their own character matrices. You could actually turn it into a separate table so it doesn't taking up that entire space of the figure, because there need to be additional taxa referred to on the diagram. Namely, you don't have any out groups in figure 7 so it's hard to describe any state specifically as ancestral and wor derived. Generally Holocephalans are the outgroup to elasmobranchs - right now they are presented as sister taxa with no ability to indicate derivation. Why isn't the catshark included in this diagram?

The character matrix is a fantastic idea, and we should have included it in the first place! We created Table 1 summarizing the traits and terminology at the end of the Introduction, also adding the character matrix in Fig7 as suggested, including specific fossil and extant species. For the Fig7 branching and catshark inclusion, please see above.

You can repurpose the figure captions as narrative body text. Use less narrative in the figure captions. These are your results actually, so move that text to the results section as a way to truncate and get to the point faster.

By figure captions, we assume the reviewer refers to figure legends. We like to explain figures to some degree of sufficiency in the legends, since some people do not read the main text and simply skim a manuscript’s abstract, figures, and figure legends. That said, we did reduce the wording, as requested.

More specific comments about semantics are listed here:The abstract starts negative and doesn't state a question although one is referenced. Potential revision - "Comprehensive examination of mineralized endoskeletal tissues warranted further exploration to understand the diversity of chondrichthyans... Evidence suggests for instance that trabecular structures are not common, however, this may be due to sampling (bring up fossil record.) We expand our understanding by characterizing the skate, cat shark, and ratfish... (Then add your current headings of the results section to the abstract, because those are the relevant takeaways.)"

We re-wrote much of the abstract, hoping that the points come across more effectively. For example, we started with “Specific character traits of mineralized endoskeletal tissues need to be clearly defined and comprehensively examined among extant chondrichthyans (elasmobranchs, such as sharks and skates, and holocephalans, such as chimaeras) to understand their evolution”. We also stated an objective for the experiments presented in the paper: “To clarify the distribution of specific endoskeletal features among extant chondrichthyans”.

In the last paragraph of the introduction, you say that "the data argue" and I admit, I am confused. Whose data? Is this a prediction or results or summary of other people's work? Either way, could be clarified to emphasize the contribution you are about to present.

Sorry for this lack of clarity, and we have changed the wording in this revision to hopefully avoid this misunderstanding.

In the second paragraph of the TMD section, you mention the synarcual comparison. I'm not sure I follow. These are results, not methods. Tell me what you are comparing directly. The non-centrum part of the synarcual separate from the centrum? They both have both parts... did you mean the comparison of those both to the cat shark? Just be specific about which taxon, which region, and which density. No need to go into reasons why you chose those regions here.. Put into methods and discussion for interpretation.

We hope that we have now clarified wording of that section.

Label the spokes somehow either in caption or on figure direction. I think I see it as part of figure 4E, I, and J, but maybe I'm misinterpreting.

Based upon histological features (e.g., regions of very low cellularity with Trichrome unstained matrix) and hypermineralization, spokes in Fig4 are labelled with * and segmented in blue. We detailed how spokes were identified in main text (lines 241-243; 252-254) and figure legend (lines 597-603).

**Reviewer #2 (Recommendations For The Authors):**
Other commentsL40: remove paedomorphism

no change; see above

L53: down tune languish, remove "severely" and "major"

done (lines 57-59)

L86: provide species and endoskeletal elements that are mineralized

no change; this paragraph was written generally, because the papers cited looked at cap zones of many different skeletal elements and neural arches in many different species

L130: remove TMD, replace by relative, descriptive, values

no change; see above

L135: What are "segmented vertebral neural arches and centra" ?

changed to “neural arches and centra of segmented vertebrae” (lines 140-141)

L166: L168 "compact" vs. "irregular". Partial mineralisation is not necessarily irregular.

thanks for pointing out this issue; we changed wording, instead contrasting “non-continuous” and “continuous” mineralization patterns (lines 171-174)

L192: "several endoskeletal regions". Provide all regions

all regions provided (lines 198-199)

L269: "has never been carefully characterized in chimeras". Carefully means what? Here, also only one chimera is analyses, not several species.

sentence removed

302: Can't believe there is no better citation for elasmobranch vertebral centra development than Gadow and Abott (1895)

added Arriata and Kolliker REFs here (lines 293-295)

L318 ff: remove discussion from result chapter

references to paedomorphism were removed from this Results section

L342: refer to the species studied, not to the entire group.

sorry, the line numbering for the reviewer and our original manuscript have been a little off for some reason, and we were unclear exactly to which line of text this comment referred. Generally in this revision, however, we have tried to restrict our direct analyses to the species analyzed, but in the Discussion we do extrapolate a bit from our data when considering relevant published papers of other species.

346: "selected representative". Selection criteria are missing

“selected representative” removed

L348: down tune, remove "critical"

Done

L351: down tune, remove "critical"

done

L 364: "Since stem chondrichthyans did not typically mineralize their centra". Means there are fossil stem chondrichthyans with full mineralised centra?

Re-worded to “Stem chondrichthyans did not appear to mineralize their centra” (lines 379)

L379: down tune and change to: "we propose the term "non-tesseral trabecular mineralization. Possibly a plesiomorphic (ancestral) character of chondrichthyans"

no change; sorry, but we feel this character state needs to be emphasized as we wrote in this paper, so that its evolutionary relationship to other chondrichthyan endoskeletal features, such as tesserae, can be clarified.

L407: suggests so far palaeontologist have not been "careful" enough?

apologies; sentence re-worded, emphasizing that synchrotron imaging might increase details of these descriptions (lines 406-408)

414: down tune, remove "we propose". Replace by "possibly" or "it can be discussed if"

sentence re-worded and “we propose” removed (lines 412-415)

L420: remove paragraph

no action; see above

L436: remove paragraph

no action; see above

L450: perhaps add summery of the discussion. A summery that focuses on the subject and the methodology of this nice study.

yes, in response to the reviewer’s comment, we finished the discussion with a summary of the current study. (lines 440-453)